# Enhancing PM2.5 prediction by mitigating annual data drift using wrapped loss and neural networks

Md Khalid Hossen[1,2,3]*, Yan-Tsung Peng[2], Meng Chang Chen[3]

**1** Social Networks and Human-Centered Computing, Taiwan International Graduate Program, Academia Sinica, Taipei, Taiwan, **2** Department of Computer Science, National Chengchi University, Taipei, Taiwan, **3** Research Center for Information Technology Innovation, Academia Sinica, Taipei, Taiwan

* kt.hossen27@iis.sinica.edu.tw

**Data Availability Statement:** 1. The main EPA data can be downloaded from the below link: https://www.kaggle.com/datasets/khalid27/dataset-epa 2. After preprocessing the EPA data, the below data

## Abstract

In many deep learning tasks, it is assumed that the data used in the training process is sampled from the same distribution. However, this may not be accurate for data collected from different contexts or during different periods. For instance, the temperatures in a city can vary from year to year due to various unclear reasons. In this paper, we utilized three distinct statistical techniques to analyze annual data drifting at various stations. These techniques calculate the P values for each station by comparing data from five years (2014-2018) to identify data drifting phenomena. To find out the data drifting scenario those statistical techniques and calculate the P value from those techniques to measure the data drifting in specific locations. From those statistical techniques, the highest drifting stations can be identified from the previous year's datasets To identify data drifting and highlight areas with significant drift, we utilized meteorological air quality and weather data in this study. We proposed two models that consider the characteristics of data drifting for PM2.5 prediction and compared them with various deep learning models, such as Long Short-Term Memory (LSTM) and its variants, for predictions from the next hour to the $64_{th}$ hour. Our proposed models significantly outperform traditional neural networks. Additionally, we introduced a wrapped loss function incorporated into a model, resulting in more accurate results compared to those using the original loss function alone and prediction has been evaluated by RMSE, MAE and MAPE metrics. The proposed Front-loaded connection model(FLC) and Back-loaded connection model (BLC) solve the data drifting issue and the wrap loss function also help alleviate the data drifting problem with model training and works for the neural network models to achieve more accurate results. Eventually, the experimental results have shown that the proposed model performance enhanced from 24.1% -16%, 12%-8.3% respectively at 1h-24h, 32h-64h with compared to baselines BILSTM model, by 24.6% -11.8%, 10%-10.2% respectively at 1h-24h, 32h-64h compared to CNN model in hourly PM2.5 predictions.

are used for model evaluation: https://www.kaggle.com/datasets/khalid27/taipei-data.

**Funding:** This document is the result of the research project funded by the National Science and Technology Council of Taiwan, under the grant NSC 109-2119-M-001-010-A. The funders had no role in study design, data collection and analysis, decision to publish, or preparation of the manuscript.

**Competing interests:** The authors have declared that no competing interests exist.

## Introduction

Air pollutants are made up of gaseous pollutants and particles that pose a threat to human health. The dispersion of a pollutant is determined by factors such as its size, origin, and ability to react with reactive oxygen. Research has shown that air quality is influenced by elevated concentrations of fine particles, specifically those with a diameter of 2.5 $\mu m$ or smaller, known as PM2.5. Studies have shown a significant impact of PM2.5 on human health [1, 2], since a recent study revealed that PM2.5 particles can easily enter and damage the human respiratory system [3]. Different researchers have proven not only respiratory diseases but also immune diseases and cardiovascular diseases related to PM2.5 have been proven by different researchers [4]. Correct predictions of PM2.5 levels help people and various sectors plan and prepare for the impacts of PM2.5 pollution, which has stimulated numerous research efforts in the field of PM2.5 prediction [5–7]. Numerous pollution sources contribute to PM2.5 in the environment. These sources include precursors of gas-phase pollutants, forest fires, agricultural burning, vehicle exhaust, industrial processes, application of agricultural fertilizers, etc. [8]. Pollution can be influenced by both local and regional sources, originating from nearby cities or even neighboring countries. Regional sources of pollution are beyond the expectations of the forecast systems, seriously altering the prediction [6]. PM2.5 levels are significantly influenced by seasonal variations, exhibiting consistent trends throughout the year. Fig 1 illustrates the weekly average PM2.5 levels in the Taipei area from 2014 to 2017, highlighting the varying patterns of PM2.5 levels in these years. In Fig 1 states the monthly change with specific dates drifts are visible.

Data drifting refers to the phenomenon in which the data distribution undergoes shifts over time. This can be observed in fluctuating PM2.5($\mu g/m^3$) levels over different years, as depicted in Fig 1. In the context of machine learning, data drift can significantly impact the performance of machine learning models [9, 10], since models are typically trained on historical data and may struggle to generalize to new, unseen, or rare distributions. It is crucial to monitor and detect data drifting and adapt models to evolving data distributions to maintain optimal model performance in dynamic environments.

PM2.5 pollution levels exhibit year-to-year variations influenced by multiple factors, such as emissions from human activities, meteorological conditions, regional and local influences, and the effects of climate change. These factors collectively contribute to the observed fluctuations in PM2.5 pollution levels over several years. Fig 1 illustrates the discernible variation in

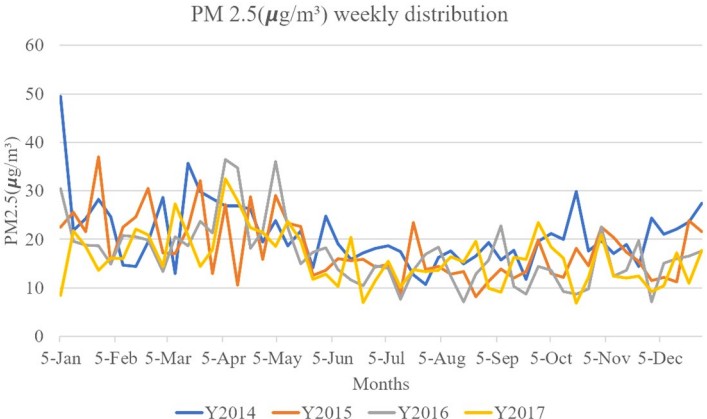

**Fig 1. Weekly average PM2.5 ($\mu g/m^3$) levels of years from 2014–2017.**

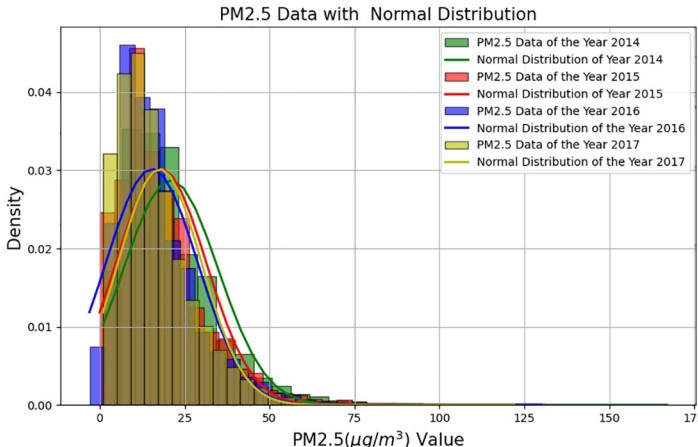

**Fig 2. PM2.5 ($\mu g/m^3$) distributions from year 2014 to 2017.**

PM2.5 levels over multiple years, from 2014 to 2017, highlighting both year-by-year fluctuations and seasonal cyclical changes.

Fig 2 shows the normal distributions of PM2.5 ($\mu g/m^3$)levels over various years, highlighting the noticeable differences between them. In Fig 2 exhibits that the yearly histogram also can check the yearly comparison as the annual phenomenon of PM2.5 levels is evident, our study adopts an annual perspective as a basis to observe and investigate the prediction of PM2.5 levels.

Various approaches can be used to examine the existence of data shifting, including statistical, machine learning, and deep learning methods. Statistical methods offer correlation analysis, which quantifies the linear association between two-time series through correlation coefficients. Autocorrelation analysis, on the other hand, examines the correlation between a time series and its previous values [11]. Another approach is machine learning, where time series classification methods, such as dynamic time warping, measure the similarity between two-time series by finding the optimal alignment between them. Support vector machines (SVMs) can also be utilized for time series analysis. Deep learning approaches, such as recurrent neural networks (RNNs), are effective in capturing temporal dependencies and patterns in time series data, making them suitable for prediction tasks [12]. Neural network models have the capability to learn data representations, recognize similarities between time series, and can further improve performance by constructing intricate neural network models.

Accurate prediction of future levels of PM2.5 is a challenging task, given the presence of numerous observable factors, such as carbon dioxide concentration, that influence PM2.5 levels. These factors, in turn, are influenced by unobservable or higher-level elements, such as government policies designed to mitigate emissions from coal burning. Constructing a comprehensive deep-learning model to capture all these intricate interactions poses a formidable challenge. As a result, this study focuses primarily on addressing the challenge of data drifting while recognizing the presence of high-level or unobservable factors that cause data drifting, which will not be examined within the scope of this paper.

Data drifting poses a well-known challenge in machine learning for future prediction [13, 14]. This study focuses on the creation of deep learning models tailored to effectively address the changing dynamics and underlying patterns in data, ensuring accurate predictions of PM2.5 levels despite the occurrence of data drifting. It is mandatory to know the pollution data scenario and which location data has drifted, most recently the research work about the

data drifting is very limited, so knowing about the datasets will help to take prerequisite steps. To increase the model performance many researchers proposed the weighted loss function add with the model training process, loss function can deal with increased model performances. The loss function can be a solution for a data drifting scenario. It is also needed to find a model that will consider the last couple of year's data to predict the hourly PM2.5 predictions. As we can collect the last couple of year's data can design a model that can transfer their knowledge to predict the next year's hourly predictions. The proposed model handles the annual data separately and employs the wrapped loss function to learn the difference between different data for the year. In our empirical study, we used data collected from Environmental Protection Agency (EPA) stations, where environmental pollutant data were collected. The primary pollutant of interest was PM2.5. Furthermore, these EPA stations measured various pollutants, including, but not limited to, PM10, CO, CO2, NO, SO2, and others. The study incorporated data from 18 stations located in the Taipei area of Taiwan. In this study, we have evaluated several baseline models such as BILSTM, LSTM, CNN, and LSTM-attention to evaluate their performance for hourly PM2.5 predictions which are widely used for time series prediction and evaluation with a proposed loss function.

The contributions of this work are as follows. In this paper, we initially utilize several formal metrics, including Maximum Mean Discrepancy, J-S divergence, and Pearson correlation, to detect occurrences of data drifting in the PM2.5 levels of the Taipei area between 2014 and 2018 to identify both drifting areas. Moreover, it has been proposed a loss function that can improve the model performance on data training time in the baseline models and proposed models to solve the data drifting scenario. Finally, propose two Convolutional neural network (CNN) based models called BLC and FLC model to predict the next hourly prediction focused on solving the data drifting scenario on PM2.5 data.

## Previous work

### Time series analysis for PM2.5 prediction

Many scholars have proposed various prediction models for forecasting PM2.5 levels [15, 16], and artificial intelligence models have demonstrated remarkable performance in PM2.5 predictions in recent studies [17, 18]. In the realm of air quality prediction, neural networks have become widely adopted due to their impressive performance [19–21]. Presently, deep learning models have achieved remarkable results in various prediction tasks, including PM2.5 predictions. The learning on the relation between the PM2.5 concentrations and climate variables and also considered the future context data sequences for model training, also encountering the correlation between air pollutants and climate variables [22]. They did not work on yearly data drifting or changing scenarios, moreover, they focused on future context but did not consider the previous year's data. As RNN-based networks did not consider how the model loss function can work efficiently to solve those issues we propose a model with a loss function and also show the way to know calculate the data drifting in the datasets. Notable models applied in this context include Bidirectional LSTM (BILSTM) based models [22, 23], Long Short-Term Memory (LSTM) models [24, 25], Convolutional Neural Network-Long Short-Term Memory (CNN-LSTM) based models [26], and CNN-bidirectional LSTM (biLSTM) based models [27].

Kavianpour et al. [28] proposed a CNN-based model that combines the spatial and temporal information that can do it by combining CNN and LSTM models. One single model has some difficulties in properly handling time series data. Li et al. [29] stated a hybrid CNN-LSTM model has been developed to improve the accuracy of forecasting PM2.5 concentration levels. This model combines a Convolutional Neural Network (CNN) with a Long

Short-Term Memory (LSTM) network to leverage the strengths of both architectures. The model [29] takes multivariate time series data as input and generates multi-step predictions for a single target variable for PM2.5 concentration. Zhu et al. [30] elaborated that the nearby station with the target station for hourly PM2.5 forecasting. The paper [30] proposed a parallel multi-input 1D-CNN-BILSTM model by combining data from both target stations and nearby monitoring stations data together. Various deep learning models are applied differently both have gotten good results for PM2.5 forecasting. Zhu et al. [31] proposed an attention-based CNN-LSTM multilayer neural network to predict PM2.5 concentration for the next 72h. This model can extract short-term and long-term temporal features simultaneously.

Among those works, Kristiani et al. [32] introduced a PM2.5 forecasting model employing the Long Short-Term Memory (LSTM) sequence-to-sequence method. Additionally, Kristiani et al. [32] employed various statistical techniques, including correlation analysis, XGBoost, and chemical methods, to identify crucial features for their model.

Despite the inherent limitations and the reduced precision of Himawari-8 predictions compared to MODIS, the Kristiani et al. [32] delved into satellite-based estimation of ground-level PM2.5, evaluating model performance through metrics such as RMSE and R values. The findings revealed that changes in PM2.5 were evident at hourly rates and that specific locations experienced significant fluctuations. BILSTM showed better performance than LSTM has shown by Siami-Namini et al. [33], particularly for time series data. In particular, Siami-Namini et al. [33] focused mainly on the BILSTM and LSTM models. Many time series analysis studies frequently neglect the issue of data shifting, a factor that can exert a considerable impact on their predictive performance. For time series prediction, LSTM models have impediments affecting evaporating gradients and simple feature loss in long sequences. Vaswani et al. [34] discussed that attention mechanism and showed many ways to solve the time series prediction problems. The attention mechanism used the softmax function to represent the probability distribution function and also the attention functions as a vector. Attention is used to take all input data information and predict the output from the data information. The paper [34] did not discuss the data scenario and also didn't focus on identifying the data drifting scenario and model loss function. Wen et al. [35] magnified that LSTM-based model with attention mechanism into the encoder-decoder structure for time series prediction and compared with baseline models. In the paper [35] did not disclose the data drifting scenario and also didn't concern model loss function to increase the model performances.

## Data shifts detection

Rabanser et al. [36] addressed the issue of dataset shifting using machine learning methods. Rabanser et al. [36] work delves into methodologies for detecting data set changes and pinpointing examples using a two-sample-based testing approach. Specifically, the study examines situations where the target data are disproportionately represented and attempts to identify shifts in the data distribution. The paper introduces multiple dimensionality reduction (DR) techniques and compares their performance. In study Rabanser et al. [36], they used the maximum mean discrepancy (MMD) technique, a popular kernel-based method for multivariate two-sample testing, to distinguish between the probability distributions of two datasets. In addition, they employed the Kolmogorov-Smirnov (KS) test to compare two different continuous distributions. To determine statistical significance, Rabanser et al. [36] obtained the P value through a permutation test on the resulting index. Fang et al. [14] explored a robust importance weighting (IW) technique designed to tackle dataset shifting through a two-step process. In the initial step, weight estimation helps to gauge the training density ratio. Subsequently, in the second step of weighted classification, the classifier undergoes training using

the weighted training data. Fang et al. [14] introduced dynamic importance weighting for deep learning, demonstrating its effectiveness in detecting covariate and class-prior shifts, as well as label noise. Gretton et al. [37] investigated statistical methods for assessing whether two samples originate from the same distribution. They introduced two categories of statistical methods: one focusing on test statistics and the other on the asymptotic distribution of the statistics. The paper, Gretton et al. [37] introduced the Reproducing Kernel Hilbert Space (RKHS) concept for testing statistics between two distributions. Pearson Correlation(PC) methods are also used to find the difference between two variables, it is the first formal that can measure the correlation and is widely used to find the relationships [38] between the data variables.

## Transfer learning for data shifting

Wei et al. [39] explored the application of transfer learning to leverage knowledge from a source city and apply it to a target city. The prerequisite for successful transfer learning is that the source city must have sufficient knowledge that can be transferred to the target city. Wei et al. [39] focused on solving the problem where the new city developed and does have not enough infrastructure, and in some cases, they don't have enough pollutant sources from where the sources of pollutant exist in the area. Wei et al. [39] transferred the knowledge from one enriched city called Beijing to transfer knowledge to another underdeveloped city called Baoding. They have needed source city and target city needed to predict but in our proposed model it is required any source or target city. Here they did not discuss the data drifting issues how the drifting can be identified and what is the possible solution for this type of problem. Our proposed model can predict from the last couple of year's datasets and transfer the yearly knowledge to the next year's prediction.

On the other hand, Cai et al. [40] pointed out that most domain adaptation works assume equal probability distributions for both source and target domains, leading to excellent performance for non-time series data. However, this assumption does not hold for time series data, particularly when data-shifting problems are present. Existing methods that rely on RNN-based models assuming equal domain distributions, do not effectively handle time series data due to this reason. To address this limitation, Cai et al. [40] proposed a CNN attention-based model designed to make hourly predictions from time series data, thereby improving the prediction performance and overcoming domain adaptation challenges specific to time series data.

## Weights on multi output loss functions

In the paper, Golik et al. [41], investigate the error criteria that occurred in the training session of the Artificial neural networks. They prove that the randomly initialized weight does not converge to the good local minima in the training of Artificial Neural Networks for the mean squared error rate. The randomly initialized weights and environment were compared, and cross-entropy gets better local optima compared with squared error rates. The squared error rate is getting worse at local minima where the gradient vanished and also found that no experiment results for classification tasks. In the paper, Kendall et al. [42] examined that the performance of the model depends on each relative weighting for each task, and tuning on hand is quite impossible and also very expensive in the machine learning training process. Kendall et al. [42] proposed a multi-task deep learning that weights multiple loss functions that focus on the homoscedastic uncertainty of each task. Kendall et al. [42] proposed a new novel and multi-task that can easily learn the various classification and regression losses by solving the homoscedastic tasks and designing a semantic segmentation architecture but it didn't consider how it can work more perfectly in a single optimal weighting for all tasks. Kendall et al. [42]

didn't propose a unified model for PM2.5 hourly prediction to consider with data drifting which can make more good results for time series data. The work proves the importance of loss weighting in multi-task deep learning and shows how one can get a better performance equivalently separated trained models.

## PM2.5 data drifting

Data drifting is a significant concern that can affect the accuracy of machine learning predictions. In this section, we will explore the phenomenon of data drifting of PM2.5 levels, which is evident in the changing data distribution over different years. Data drifting is identified when there is a dissimilarity in the statistical characteristics of feature distributions between two separate time series separated by a specific time interval.

Consider $x_t$ as the observable features at time t and $y_t$ as the corresponding concentration level of PM2.5. Here, $[y_i, i = a, \ldots, b]$ represents the time series of $y$ spanning from time a to b. $Y_1$ and $Y_2$ are two-time series that $Y_1 = [y1_1, y1_2, y1_3, \ldots, y1_t]$ and $Y_2 = [y2_1, y2_2, y2_3, \ldots, y2_t]$. Let $P(Y)$ be the distribution of $Y$. $Y_1$ and $Y_2$ have a time-drifting phenomenon if

$$P(Y_1) \napprox P(Y_2) \tag{1}$$

where $\napprox$ is a data drifting inequation, which will be explained in the next chapter. Note that in this study, the length of a time series is set at one year. Thus, $Y_i$ is the time series of PM2.5 concentrations over a whole year.

In Table 1, Y means the year, accordingly Y2014 means the year of 2014, PM2.5 data.

Analysis of data from different years at the same stations presents a challenge in identifying the root causes of these fluctuations. Table 1 shows the annual variations in mean, standard deviation, minimum, and maximum values from 2014 to 2018. The table provides further information on this phenomenon, highlighting the consistent changes in PM2.5 values observed from year to year.

## Verifications of data drifting

In this section, we will use various measurement techniques, including Kullback-Leibler divergence, maximum mean discrepancy, and Pearson's correlation, to assess the disparities in the distributions of PM2.5 levels across different years, as described in Eq 1. In this work, we have used multiple methods to find out the data drifting techniques as different methods are adopted with special characteristics to handle the data drift. However, data drift can be caused by different factors such as the data collection process and user behavior and sometimes choosing drifting techniques are very crucial. Moreover, where real-time monitoring needed there different data drifting techniques are required.

**Table 1. The mean and std, min and max values of PM2.5 from year 2014–2018.**

|  | Y2014 | Y2015 | Y2016 | Y2017 | Y2018 |
|---|---|---|---|---|---|
| Mean | 20.94 | 18.08 | 16.92 | 16.4 | 14.89 |
| Std | 13.88 | 13.23 | 12.09 | 11.35 | 9.942 |
| Min | 1 | 0 | -3 | 1 | 1 |
| Max | 167 | 131 | 130 | 122 | 109 |

## Divergence measures

The Kullback-Leibler divergence is defined as the measure of the disparity between the distributions of continuous random variables. Let X and Y represent two continuous random variables, and p(X) and p(Y) denote their distributions. Then Kullback-Leibler divergence [43] is defined as:

$$KL(X||Y) = \int_x p(x) log \left[ \frac{p(x)}{q(y)} \right] dx \tag{2}$$

To assess the data distance between two data distributions, the Jensen-Shannon (JS) divergence is often employed, derived from the Kullback-Leibler (KL) divergence. It is defined as:

$$JS(X||Y) = \frac{1}{2} KL(X||\frac{X+Y}{2}) + KL(Y||\frac{X+Y}{2}) \tag{3}$$

In both KL Divergence and JS Divergence, smaller values indicate that the two distributions are more similar. In this study, each year is associated with its own yearly PM2.5 data distribution, and we compared the data distribution of one specific year with that of another year with the same EPA stations. In Fig 3, JS divergence is used to compare how the yearly data have evolved in the past five years. JS divergence calculates the P value that involves comparing the JS divergence for every pair of years within the most recent 5 years for each EPA station. From Fig 3, it can be seen that each station has a different degree of data drifting from Eq 3. For example, the second station on the left (Cailiao) has more obvious data drifting than the first station (Banqiao).

In Fig 4 included the highest four stations P value by JS divergence, it is presented by heatmap diagram so can clearly understand the value differences, this value is calculated in the time of yearly comparison, here in Datong and Guting has maximum values.

## Maximum mean discrepancy

The Maximum Mean Discrepancy (MMD) is a kernel-based statistical test that is used to determine whether two given distributions are identical [36, 37]. The maximum mean discrepancy allows us to differentiate between p and q where the mean embeddings are $\mu_p$ and $\mu_q$ are also reproducing kernel Hilbert space

$$MMD(F, p, q) = \left\| (\mu_p - \mu_q) \right\|_F^2 \tag{4}$$

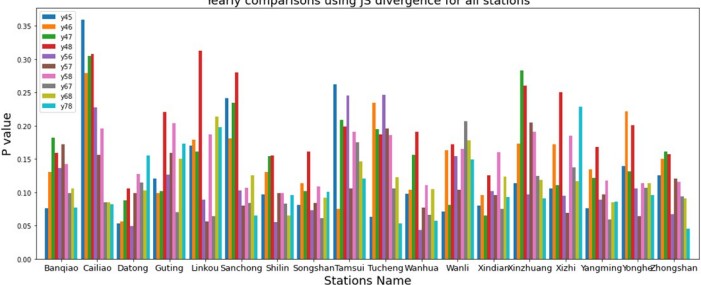

**Fig 3. Yearly comparisons using JS divergence for all stations.**

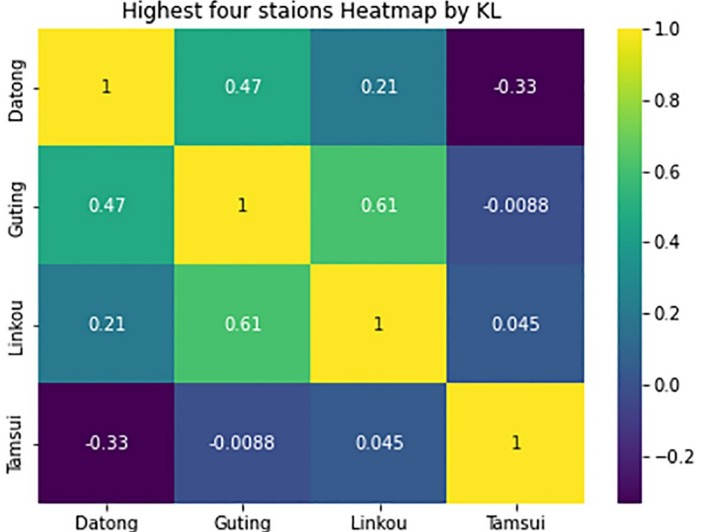

**Fig 4. Yearly comparisons using JS divergence for four stations by heatmap.**

The value of MMD close to zero indicates that the two distributions are similar. Given samples from both the X and Y distributions of length $m$, MMD can be calculated as follows:

$$MMD^2 = \frac{1}{m*(m-1)} \sum_{i=1}^{m} \sum_{j \neq i}^{m} k(x_i, x_j) + \frac{1}{m*(m-1)}$$

$$\sum_{j=1}^{m} \sum_{i \neq j}^{m} k(x_i, x_j) + \frac{1}{(m*m)} \sum_{i=1}^{m} \sum_{j=1}^{m} k(x_i, x_j)$$

(5)

In Fig 5, MMD is used to assess distance and observe how yearly data have evolved over the past 5 years. It involves comparing the MMD values for every pair of years within the most recent 5 years for each EPA station. From Fig 5, it can be seen that each station has a different degree of data drifting. For instance, the second station on the left (Cailiao) has more obvious data drifting than the first station (Banqiao), have got those results and Fig 5 from Eq 5. In Fig 6 included the highest four stations of P value by MMD, it is presented by heatmap diagram so can clearly understand the value differences, here Wanli and Xizhi stations has maximum P value.

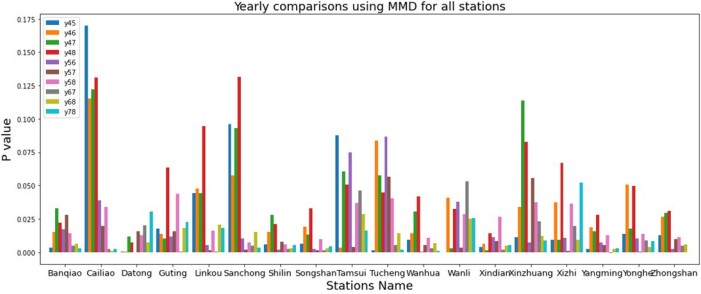

**Fig 5. Yearly comparisons using MMD for all stations.**

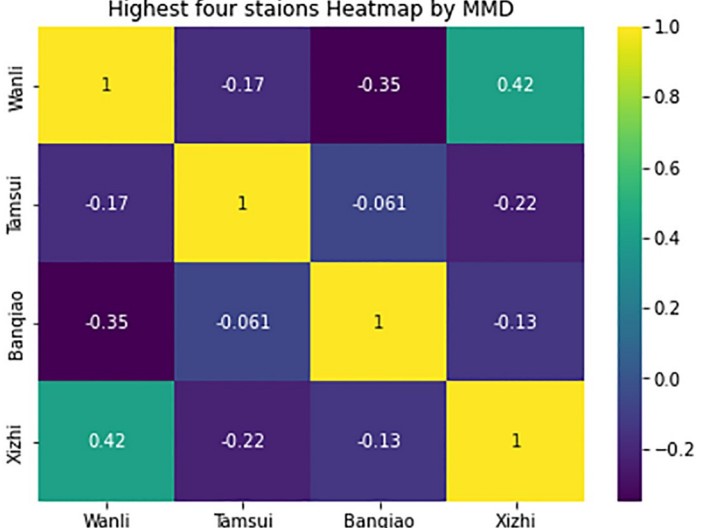

**Fig 6. Yearly comparisons using MMD for four stations by heatmap.**

## Pearson's correlation(PC)

Pearson's correlation, known as the Pearson correlation coefficient (r), is a statistical measure used to quantify the linear relationship between two continuous random variables. The Pearson's correlation produces a score between -1 to +1. Two distributions with Pearson's correlation close to +1 are considered highly similar, uncorrelated when close to zero, and inversely correlated when near -1. For two distributions X and Y, their Pearson's correlation [44] is calculated as below:

In Fig 7 included the highest four stations P value by PC, it is presented by heatmap diagram so can clearly understand the value differences, Datong and Wanli has maximum values.

$$\frac{\sum(x_i - \bar{x})(y_i - \bar{y})}{\sqrt{\sum (x_i - \bar{x})^2}\sqrt{\sum (y_i - \bar{y})^2}} \tag{6}$$

In Fig 8, Pearson's Correlation is employed to assess the extent of change in the yearly data over the past 5 years. When the PC values are near 0, it indicates a greater degree of drifting compared to the other year. For instance, in the case of the Banqiao station, the comparison between Y16 and Y18 exhibited a higher degree of drifting in comparison to any other two years.

## P values of measurements

In this subsection, we will calculate the scores of P value of the JS divergence, MMD, and Pearson correlation for the years 2014 to 2018 to demonstrate that the PM2.5 distributions at the same stations exhibit differences. Fig 9 depicts the P values obtained from JS divergence, Pearson's correlation, and MMD measurements, facilitating a comparison of the discrepancies of different stations. Note that while JS divergence, Pearson's correlation, and MMD measurements may all indicate some form of data-shifting phenomenon, they are not compatible with

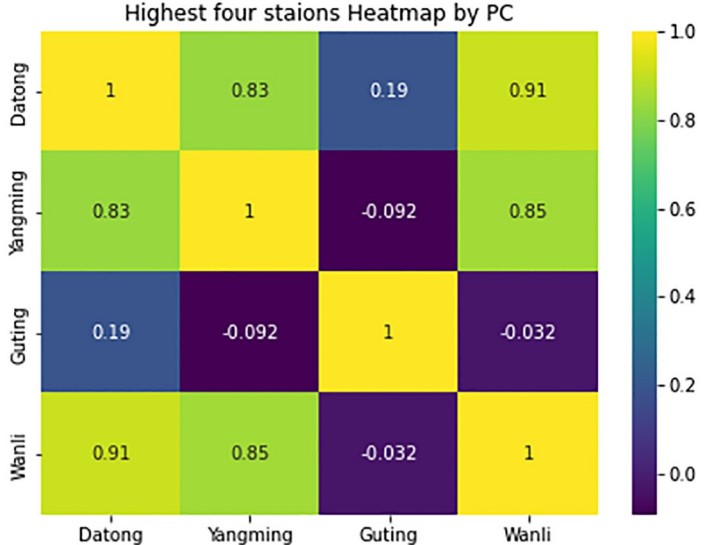

**Fig 7. Yearly comparisons using PC for four stations by heatmap.**

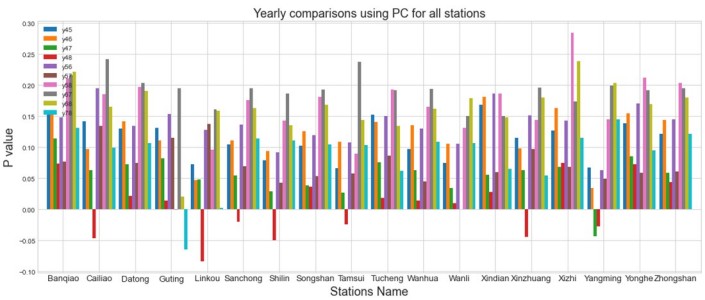

**Fig 8. Pearson correlation comparison for all stations.**

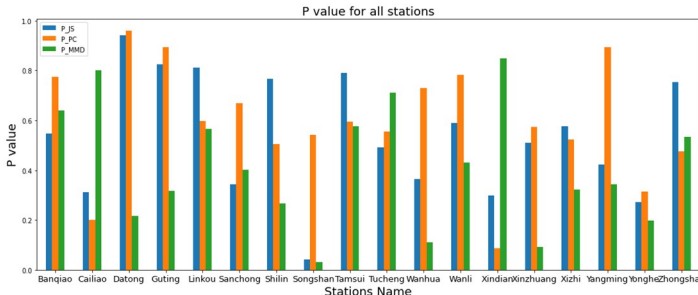

**Fig 9. P values for different measurements.**

each other. Therefore, we examine the P score of each measurement separately. We also compiled a list of stations with the highest and lowest p scores, together with their corresponding values for each measurement. Here, P-JS, P-MMD, and P-PC mean P value after JS divergence, MMD, and PC techniques.

**Table 2. Four stations of highest P values.**

| Four Stations of Highest P values | | | | | |
|---|---|---|---|---|---|
| Stations | P-JS | Stations | P-MMD | Stations | P-PC |
| *Datong* | 0.94 | *Wanli* | 0.78 | *Datong* | 0.96 |
| *Guting* | 0.82 | *Tamsui* | 0.75 | *Yangm.* | 0.89 |
| *Linkou* | 0.81 | *Banqiao* | 0.61 | *Guting* | 0.89 |
| *Tamsui* | 0.80 | *Xizhi* | 0.42 | *Wanli* | 0.78 |

**Table 3. Four stations of lowest P values.**

| Four Stations of Lowest P values | | | | | |
|---|---|---|---|---|---|
| Stations | P-JS | Stations | P-MMD | Stations | P-PC |
| *Songs.* | 0.04 | *Song.* | 0.017 | *Xindian* | 0.07 |
| *Yonghe* | 0.27 | *Guting* | 0.02 | *Yonghe* | 0.19 |
| *Xindian* | 0.29 | *Wanhua* | 0.05 | *Zhonghe* | 0.32 |
| *Cailiao* | 0.31 | *Xindian* | 0.07 | *Cailiao* | 0.34 |

Notably, we observed that some stations exhibited very similar P values. Our analysis using these techniques revealed stations with both maximum and minimum P values, as discussed in [45]. P value can be different by factoring in many cases such as sample size, variability in data, hypothesis data and other factors. Therefore, different analyses can yield different P values, even when examining the same data [46–48]

In Table 2, the EPA stations listed exhibit the highest P values for each measurement. Stations with notably high P values relative to others indicate a significant degree of data drifting. On the contrary, the stations with the lowest P values over the past five years indicate minimal data drifting. As indicated in Table 2, the Datong, Guting, and Wanli stations consistently display high P values for all three statistical measurements, highlighting a trend of substantial data drifting in these stations. In Table 3, the EPA stations have the lowest P value by applying the different statistical techniques listed. In Table 3, Songshan, Yonghe, and Xindian have the lowest data P values and are considered as the lowest data drifting.

## Wrapped loss function

The purpose of a loss function in a neural network is to quantify the disparity between the prediction and the ground truth in the training data. Imagined the trained data by A=(X,Y), where f(x)is a learned function with the weight set W. A well-designed loss function plays a crucial role in optimizing the neural network model for the given training data. The least-square loss function, based on the Gauss-Markov theorem [49], is widely used in neural network training.

$$l_{ls} = \sum_{i=1}^{c} (y_i - f(x_i))^2 \tag{7}$$

where $c$ is the length of time series. A weighted loss function incorporates a trainable weight denoted as $o_i$, into the loss functions.

$$l_{wl} = \sum_{1=1}^{c} o_i (y_i - f(x_i))^2 \tag{8}$$

Its gradient descents are as follows.

$$\frac{\partial l_{wl}}{\partial o} = \sum_{1=1}^{c} (y_i - f(X_i))^2 \tag{9}$$

$$\frac{\partial l_{lo}}{\partial w} = \sum_{1=1}^{c} 2o_i \frac{\partial f(x_i)}{\partial w}(y_i - f(x_i)) \tag{10}$$

**Algorithm 1** Wrapped loss training algorithm
```
1: Start o = 1, w = 0 and t = 0
2: repeat
3:   Calculate g_{o_i} ← ∂l_w/∂o_i
4:   ∀i, o_{i,t+1} ← o_{i,t} − α*g_{o_i}
5:   Calculate g_{W_i} ← ∂l_w/∂o_{W_i}
6:   ∀i, W_{i,t+1} ← W_{i,t} − α*g_{W_i}
7:   t ←t+1
8: until Converge
```

Although the weighted least square error function can alleviate the data drifting problem, weights can quickly approach zero from the partial differentiation of the loss function [50]. This problem can be solved by introducing a particular loss function, called the wrapped loss function, that directs the learning process. First, a normal distribution noise with a mean of 0 and standard deviation of $\epsilon_i$ is added to the predicted output to ensure that the gradient descent can be nonzero, as defined below.

$$\tilde{y}_i = f(x_i) + \epsilon_i \quad \epsilon_i \sim N(0, \sigma_i^2) \tag{11}$$

Then the wrapped loss function $l_w$ is defined as below [50].

$$l_w = \sum_{1=1}^{c} o_i(\tilde{y}_i - f(x_i))^2$$

$$= \sum_{1=1}^{c} (o_i l_i + \log o_i^{-1}) \tag{12}$$

The training process of the wrapped loss function is presented in Algorithm 1 In the algorithm, $l_w$ denotes the wrapped loss function, x denotes data input, W denotes model parameters, o denotes weighted parameters, g denotes gradient descent, t is the number of epoch, and $\alpha$ is the learning rate.

## Empirical study of PM2.5 prediction

### Data collection and preprocessing

In this work, the yearly data from 2014–2018 are used to evaluate the base model and proposed model. In the model evalutiaon, it is used 2014–2017 yearly data for model training and yearly data 2018 for model testing. The air quality datasets have been collected from Taiwan EPA websites (data.epa.gov.tw). The EPA website, has PM2.5, PM10, carbon monoxide (CO), nitric oxide (NO), nitrogen dioxide (NO2), nitrogen oxides (NOx), ozone (O3) and sulfur dioxide (SO2). The hourly meteorological data were collected from the Canter of Weather Bureau (CWB) website (opendata.cwb.gov.tw). The dataset, include humidity, temperature, pressure, rainfall and wind speed and direction. From the collected dataset have 25 features then choose only 10 features and for Taipei area data.

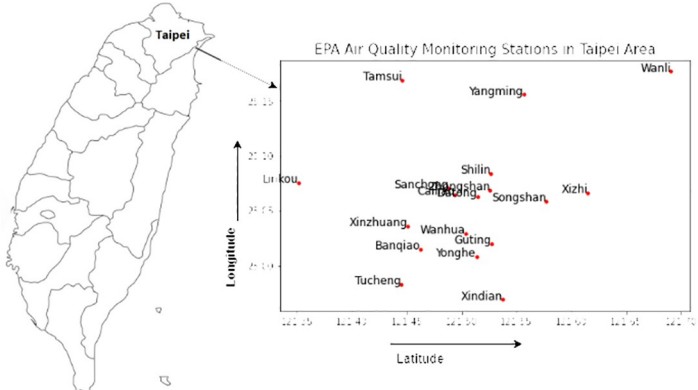

**Fig 10. The map highlighted both Taipei City and New Taipei City, indicating the locations of air quality monitoring stations.** The details can be found on the Taiwan Air Quality Monitoring Network website (https://airtw. moenv.gov.tw/ENG/Sitemap.aspx).

The Fig 10 reveals the EPA stations of the Taipei area with their longitude and latitude values. The study used a data set collected from 18 EPA stations in the Taipei area, encompassing 10 characteristics that span the years 2014 to 2017 for model training. Subsequently, the model was applied to predict PM2.5 levels for the following year, 2018. Selected features include ambient temperature, carbon monoxide (CO), nitric oxide (NO), nitrogen dioxide (NO2), nitrogen oxides (NOx), ozone (O3), PM10, PM2.5, precipitation, and relative humidity (RH). The predictions extend from the next hour to 64 hours for each of the 18 EPA stations. It has been calculated by RMSE(Root mean square value), MAE (Mean Absolute Error), MAPE (Mean absolute percentage error) matrics to evaluate the hourly prediction performances for the proposed models.

## Evelution criterian

To evaluate the baseline models and proposed model performances we have applied different methods such as root mean square error (RMSE), mean absolute error (MAE), and mean absolute percentage error (MAPE) by calculating their values. The RMSE, MAE and MAPE are calculated as shown in below equations:

$$RMSE = \sqrt{\frac{1}{n}\sum_{n}^{i=1}(A'_i - P_i)^2} \tag{13}$$

$$MAE = \frac{1}{n}\sum_{i=1}^{n}|A_i - P_i| \tag{14}$$

$$MAPE = \frac{1}{n}\sum_{i=1}^{n}|\frac{A_i - P_i}{A_i}| \times 100\% \tag{15}$$

where $A_i$ is the actual value and $P_i$ is the predicted value.

## Model development

In this section, discuss the different models like some baselines models and proposed models.

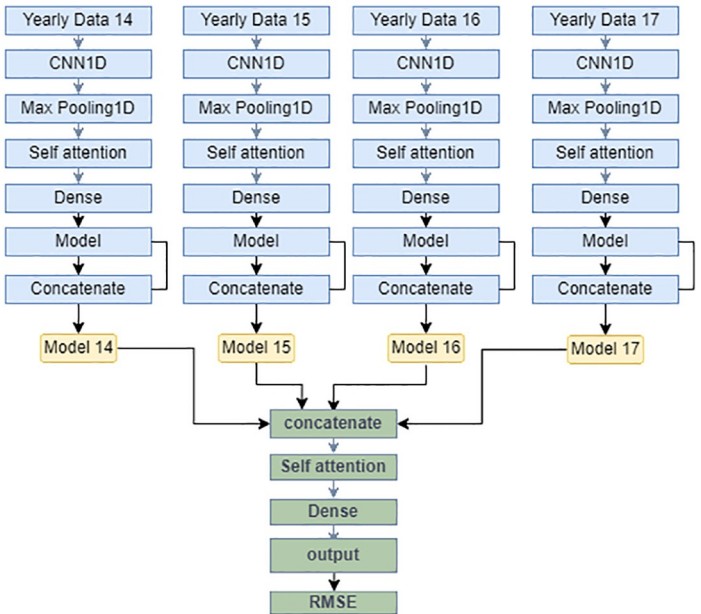

**Fig 11. Front-loaded connection model.**

**LSTM and BILSTM.** LSTM networks are widely used in time series prediction and exactly for PM2.5 hourly predictions [24, 25] and Bidirectional LSTM (BILSTM) based models [22, 23]. In our experiment, it is applied both LSTM and BILSTM models. Our work have evaluated both of those models and included their result in the result section.

**LSTM-attention.** The LSTM-attention model is also applied to the time series [51] prediction, in our work we have applied for hourly PM2.5 prediction its results are included and compared with other models.

**CNN.** In this work, applied CNN network to compare the performance for hourly prediction [30]. We have applied the attention-based model to compare the results in the results sections.

**FLC and BLC models.** Front-loaded connection model (FLC), in the first from our one yearly processed data sent to the input of CNN1D model then it goes through the Self-attention mechanism. After that, the data run through a dense model. Then four yearly data will be concatenated and sent through the attention model. Finally after training the model RMSE value calculates the model performances.

Recognizing the data drifting phenomenon in PM2.5, it is advantageous to handle PM2.5 data from different years separately. In this study, our approach involved the development of a distinct neural network model for each year, and subsequently, the outputs of all models were concatenated. The concatenated data were then fed into the back-end neural network. Two different designs were used: the first method called the front-loaded connection (FLC) model (Fig 11), used a heavy model to learn the representation for each year, supplemented by a lighter model to predict the PM2.5 level using concatenated data.

On the contrary, the second method, known as the back-loaded connection (BLC) model with a wrapped loss (Fig 12), adopted a lighter model for each individual year while employing a heavy model for concatenated data.

In the BLC model, the four years of data used as input data are concatenated together and then run with the CNN1D model. Then the output of the CNN1D model runs through

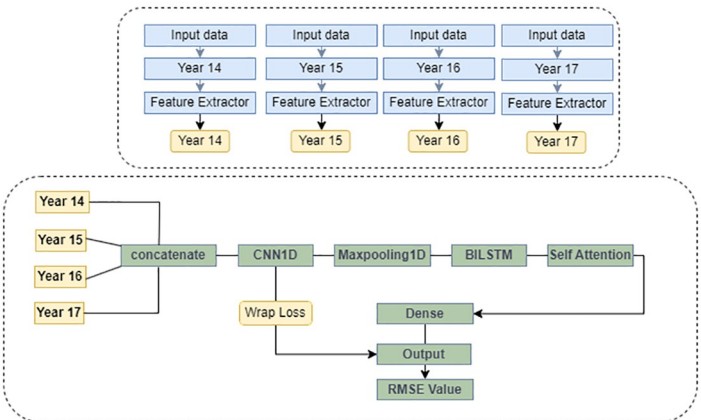

**Fig 12. Back-loaded connection model with wrap loss.**

BILSTM and runs with self-attention models. In this model for more accurate results, our proposed loss function is also applied to get good hourly PM2.5 prediction results. The details results description are described in the results sections. Furthermore, the proposed method incorporates the Wrapped Loss function to better guide the model training.

## Prediction results

Fig 11 presents the front-loaded model, where each year is handled independently using a model that incorporates CNN and self-attention mechanisms. The concatenated results then undergo an attention mechanism to make predictions. In Fig 13, the RMSE($\mu g/m^3$) values of various comparison methods are presented, which include CNN, LSTM, BILSTM, LSTM-AT the front-loaded connection model (FLC), and the back-loaded connection model (BLC). Additionally, some models that use the wrapped loss function (instead of traditional RMSE) are included for comparisons, such as WLSTM, WBILSTM, WLSTM-AT the wrapped back-loaded connection model (WBLC), and the wrapped front-loaded connection model (WFLC). The X-axis represents the prediction hours, ranging from the next hour to 64 hours. The results clearly demonstrate that both the front-loaded and back-loaded models proposed outperform other models, and the wrapped model exhibits superior performance compared to the one without the wrapped loss function.

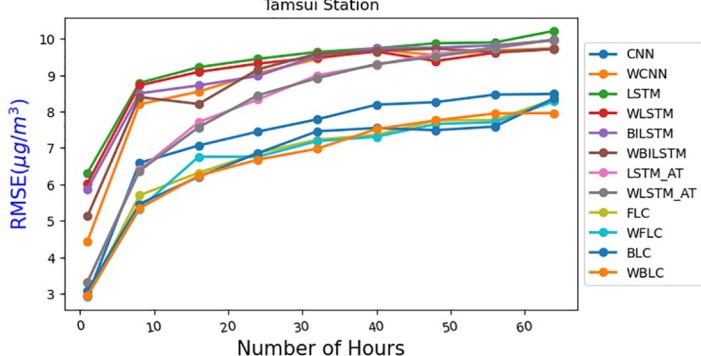

**Fig 13. RMSE ($\mu g/m^3$) of different models for Tamsui.**

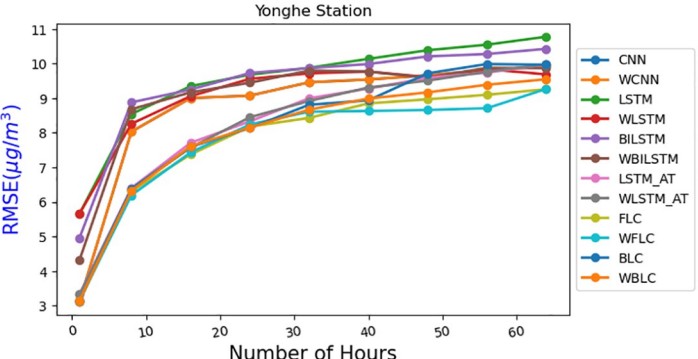

**Fig 14. RMSE ($\mu g/m^3$) of different models for Yonghe.**

In Fig 13, we selected the Tamsui station due to its high P values, as indicated in Table 2. For Fig 14, the Yonghe station was chosen for its low P values, as shown in Table 3. Tamsui is situated on the river and near the seashore, experiencing noticeable data shifting year by year. In contrast, Yonghe is a densely populated area with consistently similar PM2.5 levels in different years. Interestingly, predictions for long hours tend to be more accurate at stations with high P values in the FLC and BLC models. The models with the wrapped loss function, such as WFLC and WBLC, also outperform their non-wrapped models, FLC and BLC, no matter their P values, except for the next one-hour prediction, which heavily relies on the PM2.5 level of the previous hour. The Fig 15 have exhibited MAE($\mu g/m^3$) for (a) Banqiao and (b) for Cailiao in both cases FLC and BLC models are performing satisfactorily. Our baseline models

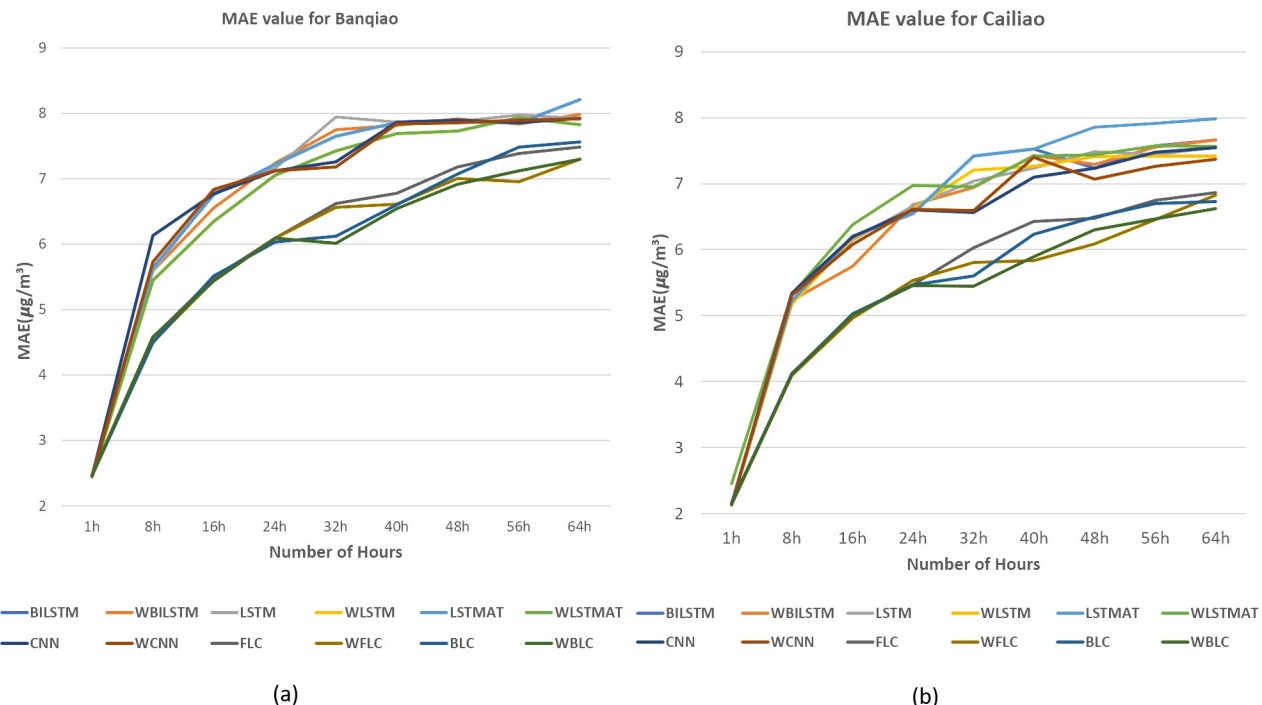

**Fig 15. MAE($\mu g/m^3$) value (a) for Banqiao (b) for Cailiao.**

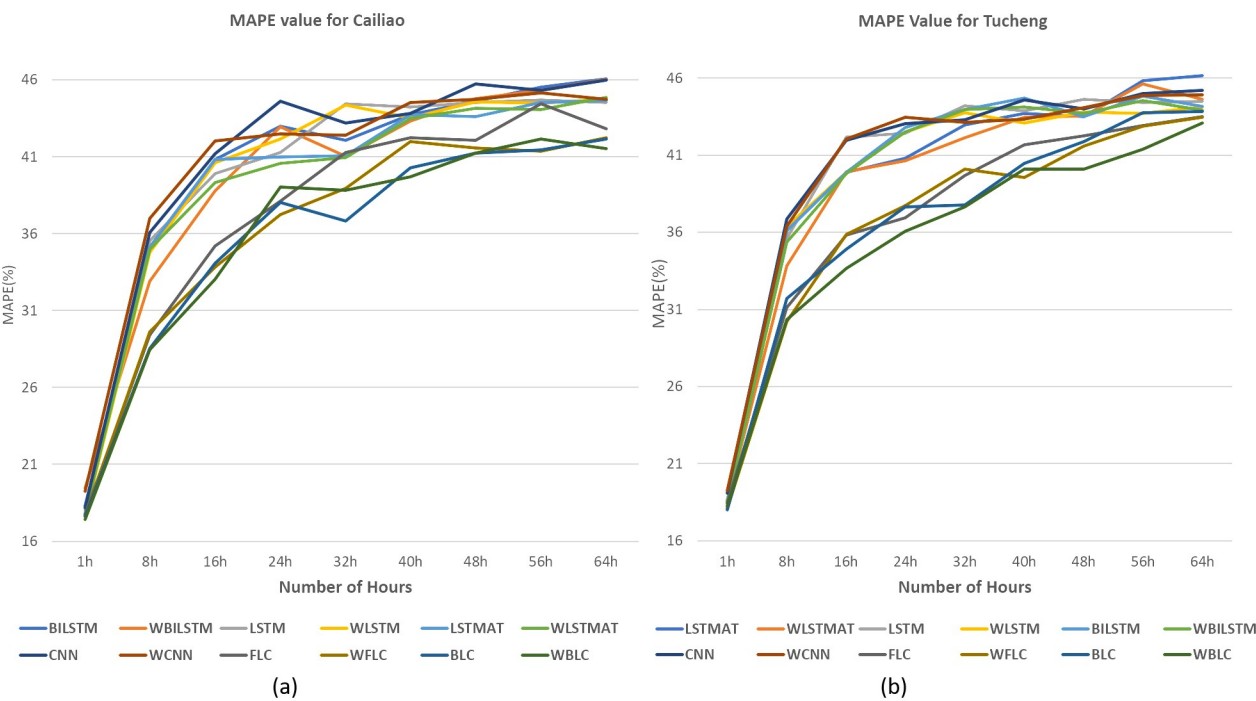

**Fig 16. MAPE(%) value for (a) for Cailiao (b) for Tucheng.**

BILSTM, LSTM, and CNN, LSTM-AT performance are also shown in Fig 15, and the proposed models BLC, and FLC have differences that are shown there. Our proposed model worked perfectly when the number of hours increased and with a comparison of the baseline model with wrap loss as WLSTM-AT our proposed model BLC worked well. Fig 16 has conducted that in (a) Cailiao and (b) for Tucheng both stations their MAPE(%) value is less for the BLC and FLC models in comparison to the baselines models with LSTM, CNN, BILSTM and with their Wrap loss functions. In both stations of Fig 16, the baseline models with wrap loss are also well performed as WCNN compared with CNN. The number of hours from 48h-64h then MAPE(%) value increased slightly. In Fig 17, the graph illustrates the average RMSE ($\mu g/m^3$) of all models of each station for predictions at different hours. The averages are identified BLC and FLC models are performed well among the all baselines models. Fig 17 clarify too that our proposed model BLC and FLC achieved good results compared to the traditional model. The wrap loss with BLC and FLC is also conducted with better results. It is evident that WFLC performs the best, and the models with wrapped loss functions outperform those without wrapped loss functions, such as WBILSTM perform better than BILISM.

## Evaluation

**Discussion for RMSE value.** Table 4 compares the performance of BILSTM, LSTM, CNN, and WFLC with FLC, WBLC, and BLC, with percentages indicating the overhead of the models compared to WFLC. The model WFLC comparison proved that the performance varies from the 8.3%-24.1%, 9% -26%, 10.2% -24.6% respectively BILSTM, LSTM, CNN model for 1 h to 64h prediction. WFLC exhibits superior performance compared to all three FLC, WBLC, and BLC, except for specific predictions for the next hour. In comparison WFLC with FLC models performance increased from 5%-0.83%, 2.1% -2.9% in time respectively for 8h-

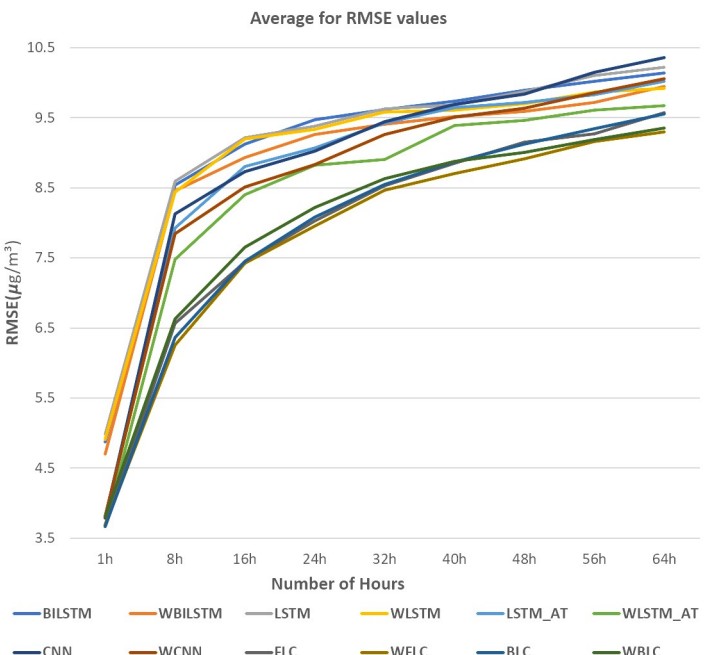

**Fig 17. Average RMSE($\mu g/m^3$) of all stations for different models.**

32h and 40h-64h, in comparison with WBLC by 3.0%-3.4%, 2.0%-0.5% for 1h-24h, 32h-64h respectively. From Figs 13–17, it is evident that the proposed wrapped loss function benefit models for PM2.5 level predictions. For example, WLSTM, WBILSTM, and WCNN, WLSTM-AT(WLS-AT) exhibit superior performance compared to LSTM, BILSTM, and CNN, LSTM-AT models.

**Discussion for MAE($\mu g/m^3$) value.** From Fig 18, it is revealed that the proposed model BLC with Wrap loss is performing well compared to all other models like WBILSTM is better compared to BILSTM. It is indicated that the LSTM, BILSTM, and CNN, LSTM-AT models are performing well but not well compared to our proposed model BLC models. We have applied LSTM attention but it's not performing well as compared to our proposed BLC and FLC model. The details results for Sanchong station are given in Table 5 where it is observed that the MAE values increase when the number of hours is increasing. Compared with all

**Table 4. The average model performances for proposed models for RMSE.**

| RMSE Increment Compared with WFLC | | | | | | |
|---|---|---|---|---|---|---|
| **Hours** | **BILSTM** | **LSTM** | **CNN** | **FLC** | **WBLC** | **BLC** |
| *1h* | 24.1 % | 26% | 24.6% | −8.1% | 3.0% | −0.5% |
| *8h* | 26.7% | 27.2 % | 23% | 5.0% | 6.1% | 1.8% |
| *16h* | 18.6% | 19.4% | 15 % | 2.7% | 3.1% | 0.4% |
| *24h* | 16 % | 15.2% | 11.8% | 1% | 3.4% | 1.6% |
| *32h* | 12% | 12% | 10 % | 0.83% | 2.0% | 1.0% |
| *40h* | 10.5 % | 10 | 9% | 2.1% | 1% | 2.0% |
| *48h* | 9.8% | 9.7% | 9.3% | 0.9% | 13% | 2.4% |
| *56h* | 8.6% | 9.3% | 9.8% | 1.2% | 0.3% | 2.0% |
| *64h* | 8.3% | 9% | 10.2% | 2.9% | 0.5% | 2.8% |

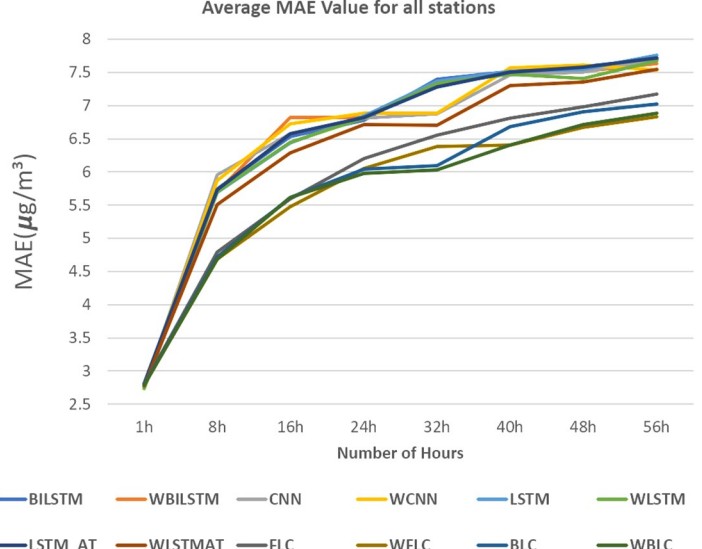

**Fig 18. Average MAE($\mu g/m^3$) comparisons for all stations.**

models, the BLC with Wrap loss and FLC with wrap loss are performing well. MAE($\mu g/m^3$) value for Sanchong station shows that our proposed model is well performed and WBLC is the best among them.

**Discussion for MAPE percentage.** From Fig 19, it is displayed that the MAPE percentage for our proposed models is quite low compared to other models. It is clearly observed that our proposed model WBLC and FLC are performed better than the baseline model in all stations. In comparison with all other models wrap loss with BLC has minimum MAPE for 64h prediction. From Table 6 it is exhibited that the MAPE percentages increase accordingly after increasing the number of hours. Table 6 includes results for BILSTM, WBILSTM, LSTM, WLSTM, LSTM-AT, WLSTM-AT, CNN, and WCNN models. Both FLC and BLC outperform these models, with the BLC model combined with the wrap loss achieving the lowest MAPE, reflecting its robust predictive accuracy across extended hours.

**Table 5. The base model and wrap loss model performance for MAE ($\mu g/m^3$)value.**

| Station name: Sanchong | | | | | | | | | |
|---|---|---|---|---|---|---|---|---|---|
| **Model** | **1h** | **8h** | **16h** | **24h** | **32h** | **40h** | **48h** | **56h** | **64h** |
| BILSTM | 3.41 | 6.62 | 7.04 | 6.97 | 7.33 | 7.65 | 7.85 | 8.11 | 8.46 |
| WBILSTM | 3.3 | 6.6 | 6.67 | 6.93 | 6.63 | 7.03 | 7.43 | 7.35 | 7.77 |
| LSTM | 3.41 | 6.59 | 6.87 | 6.92 | 6.96 | 7.29 | 7.38 | 8.37 | 8.75 |
| WLSTM | 3.16 | 6.57 | 6.8 | 6.89 | 6.93 | 7.26 | 7.39 | 7.73 | 7.83 |
| LSTM − AT | 3.41 | 6.62 | 7.04 | 6.97 | 7.33 | 7.46 | 7.52 | 8.11 | 8.62 |
| WLS − AT | 3.21 | 6.48 | 6.73 | 6.91 | 7.03 | 7.45 | 7.26 | 7.56 | 7.85 |
| CNN | 3.2 | 6.9 | 7.04 | 6.87 | 6.86 | 7.85 | 7.38 | 7.92 | 8.5 |
| WCNN | 3.18 | 6.71 | 7.41 | 6.84 | 6.89 | 7.56 | 7.24 | 7.43 | 8.23 |
| FLC | 3.18 | 5.83 | 6.62 | 7.16 | 7.5 | 7.84 | 7.62 | 7.75 | 8.36 |
| WFLC | 3.26 | 5.27 | 6.03 | 7.38 | 7.51 | 7.42 | 7.66 | 7.38 | 8.29 |
| BLC | 3.2 | 6.94 | 6.9 | 6.79 | 7.33 | 7.67 | 7.63 | 7.45 | 7.63 |
| WBLC | 3.18 | 6.55 | 7.06 | 6.88 | 7.06 | 7.21 | 7.62 | 7.26 | 7.58 |

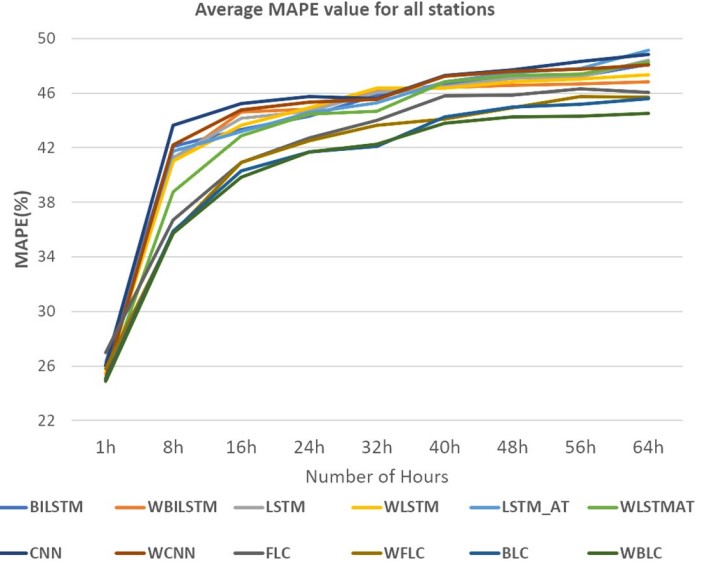

**Fig 19. MAPE (%) comparisons for all stations.**

## Conclusion

In this work, it is calculated the data drifting scenario from the last 5 years data by different statistical techniques. After getting the P value from those techniques it is mentioned the highest and lowest data drifting stations from the datasets. We have applied the traditional model for predicting the PM2.5 hourly prediction and we have applied LSTM, LSTM-AT, BILSTM, and CNN model for hourly predictions. There is a contribution to propose the wrap loss which assists the model in increasing the model performances, and it is applied to all the baseline models and models with wrap loss got better results compared with baseline models. After considering the previous year's datasets and transferring the knowledge to next year's predictions we proposed the FLC and BLC models. Among them, BLC model with wrap loss has achieved good results as demonstrated by calculating the RMSE($\mu g/m^3$), MAE($\mu g/m^3$), MAPE

**Table 6. The base model and wrap loss model performance for MAPE (%) value.**

| Station name: Tamsui | | | | | | | | | |
|---|---|---|---|---|---|---|---|---|---|
| Model | 1h | 8h | 16h | 24h | 32h | 40h | 48h | 56h | 64h |
| BILSTM | 18.66 | 34.41 | 35.1 | 37.02 | 41.34 | 40.89 | 42.19 | 42.01 | 45.44 |
| WBILSTM | 18.43 | 31.97 | 35.64 | 37.06 | 41.27 | 39.73 | 40.45 | 41.12 | 45.23 |
| LSTM | 18.58 | 36.19 | 38.32 | 37.24 | 38.34 | 40.1 | 40.07 | 43.78 | 43.66 |
| WLSTM | 18.51 | 36.66 | 37.52 | 37.44 | 38.52 | 38.73 | 39.29 | 41.42 | 41.93 |
| LSTM − AT | 18.66 | 34.41 | 37.1 | 38.02 | 40.57 | 39.78 | 39.19 | 40.01 | 41.44 |
| WLS − AT | 19.57 | 32.35 | 36.24 | 38.37 | 39.7 | 39.11 | 38.82 | 39.99 | 40.97 |
| CNN | 18.12 | 35.25 | 40.76 | 40.57 | 40.8 | 43.78 | 42.57 | 43 | 42.76 |
| WCNN | 18.52 | 35.42 | 39.97 | 40.53 | 40.46 | 42.47 | 40.83 | 40.79 | 42.05 |
| FLC | 19.33 | 29.08 | 32.45 | 35.57 | 36.62 | 37.8 | 37.89 | 39.82 | 38.66 |
| WFLC | 18.25 | 28.27 | 33.86 | 35.47 | 35.89 | 36.69 | 37.05 | 39.95 | 38.91 |
| BLC | 18.34 | 28.22 | 33.74 | 34.34 | 34.65 | 38.1 | 39.28 | 39.52 | 38.26 |
| WBLC | 18.2 | 29.17 | 32.65 | 36.12 | 35.01 | 36.4 | 39.54 | 38.42 | 37.67 |

values. It is also viewed from the average of all models from different figures with all 18 stations. The proposed model BLC and FLC are well performed in the experimental results and increased the model performances in compared WFLC in compared with LSTM from 26%-15.2%, 10%-10.2% from 1h-24h, 32h-64h respectively. In comparison with WFLC with BLC the performance increased by 1.8%-2.8% for 1h-64h respectively. Different models have been employed to predict PM2.5 levels, but most of them do not consider the data drift problem. The proposed FLC model is utilized to explore data transfer learning from the data of the previous four years. This model yields exceptional results that it transfers learning from the last few years. In this scenario, considering the distribution of yearly data allows us to showcase how the model performs when incorporating historical data. The proposed wrapped loss function facilitates learning the difference of data from different years that can be incorporated into existing models, effectively minimizing data drifting issues and enhancing prediction accuracy. In addition, the model can leverage multiple data sources, such as Airbox, Traffic Data, and CWT data sets, to make comprehensive predictions. The techniques proposed in this paper can be applied to various time series datasets. Moreover, it should be noted that when the wrapped loss function is used with dropout or batch normalization, the training process is completed even more efficiently. In the subsequent stages, different domain-based models will be explored to obtain more accurate results for comparison with our proposed models.

In the future, the proposed work can be applied to improving satellite images to solve the PM2.5 prediction cause some authors applied the satellite images for PM2.5 prediction. Shortly we will add remote transportation data so that our performance will be increased more. Soon the work can be applied to another region too for PM2.5 prediction. In limitations, the attention mechanism may have overfitting so it can hamper the output predictions. If the time series data have more drifting then the model performances can also fluctuate. The model may overfit by memorizing patterns in the training data rather than learning to make generalizable predictions based on the input data. In many cases, the CNN-based model could not understand the entire sequences of the input data, so sometimes it affects the model performances.

## Supporting information

**S1 Table. The base model and proposed model details.**
(PDF)

## Acknowledgments

I would like to thank our lab mates Asher Shao and Guo-Wei who helped me by discussing the different issues of time series datasets. I would like to extend our gratitude to the teams at the CWB and the Environmental Protection Administration (EPA) of Taiwan for providing the invaluable data utilized in our work.

## Author Contributions

**Conceptualization:** Md Khalid Hossen.

**Data curation:** Md Khalid Hossen.

**Formal analysis:** Md Khalid Hossen.

**Funding acquisition:** Meng Chang Chen.

**Investigation:** Md Khalid Hossen, Yan-Tsung Peng.

**Methodology:** Md Khalid Hossen.

**Project administration:** Meng Chang Chen.

**Resources:** Md Khalid Hossen.

**Software:** Md Khalid Hossen.

**Supervision:** Yan-Tsung Peng.

**Validation:** Md Khalid Hossen.

**Visualization:** Md Khalid Hossen.

**Writing – original draft:** Md Khalid Hossen.

**Writing – review & editing:** Yan-Tsung Peng, Meng Chang Chen.

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
