## [Decision Letter · Decision Letter 0]

10 Mar 2024

PONE-D-24-05738On the Data Shifting: Predicting PM2.5 levels by using Enhanced Deep neural networkPLOS ONE

Dear Dr. Hossen,

Thank you for submitting your manuscript to PLOS ONE. After careful consideration, we feel that it has merit but does not fully meet PLOS ONE’s publication criteria as it currently stands. Therefore, we invite you to submit a revised version of the manuscript that addresses the points raised during the review process.

We look forward to receiving your revised manuscript.

Kind regards,

Worradorn Phairuang, Ph.D.

Academic Editor

PLOS ONE

Journal Requirements:

"This document is the result of the research project funded by the National Science and 363

Technology Council of Taiwan, under the grant NSC 109-2119-M-001-010-A."

4. Please note that funding information should not appear in the Acknowledgments section or other areas of your manuscript. We will only publish funding information present in the Funding Statement section of the online submission form. Please remove any funding-related text from the manuscript.

6. Please amend either the abstract on the online submission form (via Edit Submission) or the abstract in the manuscript so that they are identical.

7. Please ensure that you refer to Figure 12 in your text as, if accepted, production will need this reference to link the reader to the figure.

8. We note you have included a table to which you do not refer in the text of your manuscript. Please ensure that you refer to Tables 4 and 8 in your text; if accepted, production will need this reference to link the reader to the Table.

**Additional Editor Comments:**

Major revision.

Reviewers' comments:

Reviewer's Responses to Questions

**Comments to the Author**

1. Is the manuscript technically sound, and do the data support the conclusions?

Reviewer #1: Partly

Reviewer #2: Partly

2. Has the statistical analysis been performed appropriately and rigorously? 

Reviewer #1: Yes

Reviewer #2: Yes

3. Have the authors made all data underlying the findings in their manuscript fully available?

Reviewer #1: Yes

Reviewer #2: Yes

4. Is the manuscript presented in an intelligible fashion and written in standard English?

Reviewer #1: Yes

Reviewer #2: Yes

5. Review Comments to the Author

Reviewer #1: The paper presents an approach to detecting scenarios of data drifting in PM2.5 levels using various strategies, and based on these findings, proposes a CNN attention-based transfer learning model for forecasting PM2.5 levels. This approach signifies an important step towards addressing the challenges posed by data drift in environmental data analysis. However, while the exploration of such ideas is intriguing and potentially impactful, there are several concerns that obscure the paper's contribution. Primarily, the abstract does not adequately reflect the methodological proposition involving CNNs with attention mechanisms and training using an alternative loss function. Instead, it focuses predominantly on the detection techniques for data drifting and their comparison with other state-of-the-art models in time series forecasting. This omission raises questions about the thoroughness with which the proposed models are presented and their potential advantages over existing methods. It is not clear if the integration of advanced neural network architectures and novel loss functions for improved accuracy in PM2.5 prediction is also a paper's contribution as the title states. Clarify.

It's also pertinent to note that the paper's engagement with the state of the art in time series forecasting for PM2.5 prediction does not explicitly mention Long Short-Term Memory (LSTM) networks with attention mechanisms. This omission could suggest that such approaches have not been widely proposed or explored within this specific domain, marking a potential gap in the literature that the current study aims to address. The introduction of CNNs with attention mechanisms tailored for PM2.5 forecasting, as highlighted in the paper, is an innovative step. However, the absence of a discussion on LSTM networks with attention mechanisms raises questions about the comprehensive coverage of existing methodologies and their comparative analysis within the study.

Furthermore, the paper references work number 33 in the state of the art section on transfer learning, which also proposes CNNs with attention mechanisms for forecasting tasks. This reference necessitates a clear delineation of how the current paper's approach diverges from the methodologies previously outlined. The distinction appears to lie solely in the specific focus on data shifting strategies and the implementation of a novel loss function. Please Clarify in the text.

In terms of the presentation of results, the paper's Figure 1 lacks clarity due to the absence of units on the y-axis. To enhance the readability and usefulness of this figure, it is advisable to not only include units but also consider revising the representation of time. Instead of using the week number of the year, incorporating reference dates that can easily pinpoint the months covered would significantly improve the ability to identify potential seasonal patterns within the data. Such a modification would make the figure more intuitive and facilitate a deeper understanding of the temporal dynamics being illustrated.

Regarding Figure 2, which presents a comparison of the time series distributions, there is a notable absence of quantitative insights into the specific differences observed between the distributions. While visual comparisons can be informative, they leave much to the reader's interpretation and assumptions about the nature and significance of the differences. To address this, the paper should include a detailed analysis of the distributions, highlighting significant quantitative differences.

For Figures 3 to 6, where the results are presented, it's critical to address the issue of readability and interpretation posed by the extensive use of grouped bars. The dense clustering of bars makes it challenging to discern patterns or draw meaningful insights from the data. The authors are advised to consider alternative visualization techniques that can facilitate a clearer understanding of the results. One such recommendation is the use of heat maps to represent the average values, and/or the standard deviations of the different metrics found across all measurement points. Heat maps could provide a visually intuitive means to identify variations and patterns in the data, highlighting areas of interest more effectively than the current bar groupings. Another suggestion is to employ a matrix format that includes references to the years and represents the distribution shapes of the metrics found (e.g., probability functions or boxplots). This approach would allow for a more nuanced comparison across different years combinations, making it easier to identify where metrics are generally higher or lower. Furthermore, it's essential that all figures include titles on their axes.

For the presentation of time series forecasting results, it's indeed beneficial to go beyond the use of Root Mean Square Error (RMSE) as the sole metric of accuracy. Including comparisons using Mean Absolute Error (MAE) and Mean Absolute Percentage Error (MAPE) is recommended, as these metrics are also widely utilized in the field and provide different perspectives on forecast accuracy.

Lastly, it's paramount to ensure clarity and consistency in the notation used throughout the paper, especially in the mathematical equations presented. Every symbol, variable, and function within these equations should be explicitly defined and described within the text.

Reviewer #2: The authors have worked on Predicting PM2.5 levels by using an Enhanced Deep neural network; and discussed the Data drift problem along with Wrapped Loss Function. Their findings have shown the validity of their work. But, to improve the paper to a level of acceptance, the following major revisions are required:

1. The last sentence of the abstract is not complete.

2. For the readers’ advantage, they need to elaborate on the point description of the proposed FLC and BLC models along with better and more comprehensive/professional-looking diagrams. In its present form, the paper looks like copy pasted from a scientific report or a thesis.

3. In the title they mention the use of Deep NN, but in the model description they mention CNN. Why they have not mentioned CNN in the title?

4. In the Model Design Section (FLC & BLC), they mentioned the conclusion along with a description (although inadequate) of the models (in lines 320-322: The results clearly demonstrate that both the front-loaded and back-loaded models proposed outperform other models, and the wrapped model exhibits superior performance compared to the one without the wrapped loss function). Is it required here?

5. A complete description of the datasets created/used should be given in its section before the Experimental setup.

6. The results of the Experiments should be illustrated in a separate Results Section.

7. Results should be discussed appropriately in a separate Discussion Section.

8. The Conclusions should be drawn as per the findings in the research work and they should be substantiated with the outcomes.

9. The organization to present the paper needs to be improved. They may read the papers to improve the presentation.

https://doi.org/10.3390/diagnostics12051134

https://ieeexplore.ieee.org/stamp/stamp.jsp?arnumber=9740199

https://ieeexplore.ieee.org/stamp/stamp.jsp?arnumber=9956807

At the end of the Introduction section, describe the organization of the paper in brief.

10. The purpose of using many statistical methods should be properly explained in the paper.

11. A comparison of the models with SOTA-published works should be given.

12. The related Work/literature review section should be more elaborate to show the substantial research gaps and accordingly the research problem statement should be mentioned after the Introduction section.

6. PLOS authors have the option to publish the peer review history of their article (what does this mean?). If published, this will include your full peer review and any attached files.

Reviewer #1: No

Reviewer #2: No

---

## [Author Response · Author response to Decision Letter 0]

8 May 2024

Academic Editors

Comment 1: (general comment) Please ensure that your manuscript meets PLOS ONE’s style requirements,

including those for file naming. The PLOS ONE style templates can be found at https://journals.

plos.org/plosone/s/file?id=wjVg/PLOSOne_formatting_sample_main_body.pdf and

https://journals.plos.org/plosone/s/file?id=ba62/PLOSOne_formatting_sample_

title_authors_affiliations.pdf

Response I have followed the the above format in my manuscript

Comment ) 2:Please note that PLOS ONE has specific guidelines on code sharing for submissions in which author generated code underpins the findings in the manuscript. In these cases, all author-generated code must

be made available without restrictions upon publication of the work. Please review our guidelines at

https://journals.plos.org/plosone/s/materials-and-software-sharingloc-sharing-code

and ensure that your code is shared in a way that follows best practice and facilitates reproducibility

and reuse.

Response Definitely, I will share the source code and so that it can reproduce.

Comment 3) Thank you for stating the following financial disclosure:

"This document is the result of the research project funded by the National Science and Technology Council

of Taiwan, under the grant NSC 109-2119-M-001-010-A." Please state what role the funders took in the study.

If the funders had no role, please state: "The funders had no role in study design, data collection and analysis,

decision to publish, or preparation of the manuscript." If this statement is not correct you must amend it as

needed. Please include this amended Role of Funder statement in your cover letter; we will change the online

submission form on your behalf.

Response This study was supported by National Science and Technology Council of Taiwan, under the grant NSC

109-2119-M-001-010-A. No additional external funding was received for this study. The funders had no role

in study design, data collection and analysis, decision to publish, or preparation of the manuscript.

Comment 4) Specific Comment 4: Please note that funding information should not appear in the Acknowledgments

section or other areas of your manuscript. We will only publish funding information present in the

Funding Statement section of the online submission form. Please remove any funding-related text from

the manuscript.

Response: I already removed from the Acknowledge sections.

I already removed from the Acknowledge sections.

Comment 5) Specific Comment 5: When completing the data availability statement of the submission form, you

indicated that you would make your data available on acceptance. We strongly recommend all authors

decide on a data sharing plan before acceptance, as the process can be lengthy and hold up publication

timelines. Please note that, though access restrictions are acceptable now, your entire data will need to

be made freely accessible if your manuscript is accepted for publication. This policy applies to all data

except where public deposition would breach compliance with the protocol approved by your research

ethics board. If you are unable to adhere to our open data policy, please kindly revise your statement to

explain your reasoning and we will seek the editor’s input on an exemption. Please be assured that,

once you have provided your new statement, the assessment of your exemption will not hold up the

peer review process.

Response: The data we collected are called EPA data and shared in the below link https://1drv.ms/f/c/

615854b8e839ca3e/EpO3bYopjRBHjEWAWlO0WtIBU0b2NPiGliOhrsdKMxRCwA?e=RfX5z7

and after preprocessing the EPA data collected only for Taipei area and the evaluation process used this datasets

https://1drv.ms/f/c/615854b8e839ca3e/EvywgMKXuJRLian31QPWT0IBVSCWtKVWc9fu4vrqaNEKCQ?

e=txnbOZ

Comment 6. Please amend either the abstract on the online submission form (via Edit Submission) or the abstract

in the manuscript so that they are identical.

Response :I have edited the abstract from the online submission.

Comment 7. Please ensure that you refer to Figure 12 in your text as, if accepted, production will need this

reference to link the reader to the figure. Response I have confirmed about figure 12 and it’s included in the

manuscript.

Comment 8. We note you have included a table to which you do not refer in the text of your manuscript. Please

ensure that you refer to Tables 4 and 8 in your text; if accepted, production will need this reference to

link the reader to the Table.

Response: I have confirmed Tables 4 and 8 so that their text is included in the manuscript.

 Reviewer 1

Comment Reviewer 1: (general comment) This omission raises questions about the thoroughness with which

the proposed models are presented and their potential advantages over existing methods. It is not

clear if the integration of advanced neural network architectures and novel loss functions for improved

accuracy in PM2.5 prediction is also a paper’s contribution as the title states. Clarify.

Response I have discussed in detail about contributions and clarified the contributions in the end of the Abstract Section.

In this work, the proposed model and the loss function are both contributions for this work to yanking

the data drifting scenario. The CNN-Attention-based model solves the data drifting issue, and the

wrap loss function also helps alleviate the data drifting problem with model training and works for the

baseline model and neural network models to achieve more correct results.

Comment 1) Specific Comment 1: However, the absence of a discussion on LSTM networks with attention

mechanisms raises questions about the comprehensive coverage of existing methodologies and their

comparative analysis within the study.

Response After getting feedback from the reviewers, it is added the LSTM attention-based model and included in the

Model Designs section. It found that their result could not compete with our proposed model. The following

sentence is added to the Results and Discussion section. We have discussed LSTM attention with their loss

functions too.

In the LSTM-attention model is also applied to the time series [43] prediction, in our work we have

applied for hourly PM2.5 prediction its results are included and compared with other models.

Comment 2) Specific Comment 2: Furthermore, the paper references work number 33 in the state of the art

section on transfer learning, which also proposes CNNs with attention mechanisms for forecasting

tasks. This reference necessitates a clear delineation of how the current paper’s approach diverges

from the methodologies previously outlined. The distinction appears to lie solely in the specific focus on

data-shifting strategies and the implementation of a novel loss function. Please Clarify in the text.

Response Line 33, I have added our work necessities and compared them with the work that has been used as a reference

outlined in [33].

They trained their model using data from the last four years and used it to predict the PM2.5 level

of Taipei City for the next year without addressing the data drifting issue. Taking advantage of

abundant data to tackle data drifting challenges through transfer learning, they introduced a CNN

attention-based model designed to make hourly predictions from time series data, 144 thereby improving

the prediction performance and overcoming domain adaptation 145 challenges specific to time series

data. This work focused on solving the problem where the new city developed and did not have enough

data, and in some cases, they did not have enough pollutant observation of pollutant sources in the

area. The authors transferred the knowledge from one enriched city Beijing to another city Baoding.

They needed the source city and a target city for transfer learning, but in our proposed model, it does

not require any source or target city. Also, they did not discuss the data drifting issues. Our proposed

model has one more advantage that it predicts from EPA data and weather data from the last couple of

years and also includes domain adaptation

Comment 3) Specific Comment 3: In terms of the presentation of results, the paper’s Figure 1 lacks clarity due

to the absence of units on the y-axis. To enhance the readability and usefulness of this figure, it is

advisable to not only include units but also consider revising the representation of time. Instead of

using the week number of the year, incorporating reference dates that can easily pinpoint the months

covered would significantly improve the ability to identify potential seasonal patterns within the data.

Such a modification would make the figure more intuitive and facilitate a deeper understanding of the

temporal dynamics being illustrated.

Response: I have added the figure which I included dates with the month. I have added the in the figure 1 according to

the reviewer’s comments. it is stated in line 23, Introduction Section

In Fig 1 states the monthly change with specific dates can easily be visible where drifts are available.

Fig 1. Weekly Average PM2.5 Levels of Years from 2014-2017

Comment 4) Specific Comment 4: Regarding Figure 2, which presents a comparison of the time series distributions,

there is a notable absence of quantitative insights into the specific differences observed between

the distributions. While visual comparisons can be informative, they leave much to the reader’s

interpretation and assumptions about the nature and significance of the differences. To address this,

the paper should include a detailed analysis of the distributions, highlighting significant quantitative

differences.

Response I have explained the data distribution difference from the figure 2. It is discussed in line 35-39

Fig 2 shows the probability distributions of PM2.5 levels over various years, highlighting the noticeable

differences between them. As the annual phenomenon of PM2.5 levels is evident, our study adopts

an annual perspective as a basis to observe and investigate the prediction of PM2.5 levels. Fig 2

shows the normal distributions of PM2.5 levels over various years, highlighting the noticeable

differences between them. In Figure 2 it is shown that the yearly histogram can also check the

yearly comparison since the annual phenomenon of PM2.5 levels is evident, our study adopts an

annual perspective as a basis for observation and investigation of the prediction of PM2.5 levels.

Fig 2. PM2.5 distributions from Year 2014 to 2017

Comment 5) Specific Comment 5: For Figures 3 to 6, where the results are presented, it’s critical to address the

issue of readability and interpretation posed by the extensive use of grouped bars. The dense clustering

of bars makes it challenging to discern patterns or draw meaningful insights from the data. The authors

are advised to consider alternative visualization techniques that can facilitate a clearer understanding

of the results. One such recommendation is the use of heat maps to represent the average values, and/or

the standard deviations of the different metrics found across all measurement points. Heat maps could

provide a visually intuitive means to identify variations and patterns in the data, highlighting areas

of interest more effectively than the current bar groupings. Another suggestion is to employ a matrix

format that includes references to the years and represents the distribution shapes of the metrics found

(e.g., probability functions or boxplots). This approach would allow for a more nuanced comparison

across different years combinations, making it easier to identify where metrics are generally higher or

lower. Furthermore, it’s essential that all figures include titles on their axes.

Response After getting the feedback from the reviewers, I have added the heat maps for each methodology and it has been

shown from Fig 3-4 to Fig 4-5 and Fig 5-6 . In-Line included numbers are 253-254,267-268,277-278.

I have added the Heat map figure so changes can be easily identified from the different statistical techniques

Fig 3. Yearly comparisons using JS divergence for all stations

Fig 4. Yearly comparisons using JS divergence for four stations by heatmap

In Fig 4 includes the highest four stations P value by JS divergence, it is presented by heatmap

diagram so can clearly understand the value differences, this value is calculated in the time of

yearly comparison, here in Datong and Guting has maximum values.

Fig 5. Yearly comparisons using Pearson Correaltion

Fig 6. Yearly comparisons using PC for four stations by heatmap

In Fig 6 included the highest four stations of P value by MMD, it is presented by heatmap diagram

so can clearly understand the value differences, here Wanli and Xizhi stations has maximum P

value.

Fig 7. Yearly comparisons using MMD for all stations

Fig 8. Yearly comparisons using MMD for four stations by heatmap In Fig ?? included the highest

four stations P value by PC, it is presented by heatmap diagram so can clearly understand the

value differences, Datong and Wanli has maximum values.

Comment 6) Specific Comment 6: For the presentation of time series forecasting results, it’s indeed beneficial

to go beyond the use of Root Mean Square Error (RMSE) as the sole metric of accuracy. Including

comparisons using Mean Absolute Error (MAE) and Mean Absolute Percentage Error (MAPE) is

recommended, as these metrics are also widely utilized in the field and provide different perspectives

on forecast accuracy.

Response Thank you for your advice. After getting the advice of the reviewers, I have added the RMSE, MAE, and

MAPE results for PM2.5 predictions. In the Evalution Criterian section I discussed its definition. And the

Results and Discussion section I discussed about its MAE and MAPE results.

To evaluate the performance of the baseline models and the proposed model, we have applied different

methods such as the root mean square error (RMSE), mean absolute error (MAE), and mean absolute

percentage error (MAPE) to evaluate their performance. We have shown their results in Results Section.

Comment 7) Specific Comment 7: Lastly, it’s paramount to ensure clarity and consistency in the notation used

throughout the paper, especially in the mathematical equations presented. Every symbol, variable, and

function within these equations should be explicitly defined and described within the text.

8Response I have carefully checked the consistency of the notation used in this paper, and also the equations.

Reviewer 2:

Comment Reviewer 2: (general comment) 1. The last sentence of the abstract is not complete.

Response There was a mistake. I have completed the first line.

Additionally, we introduced a wrapped loss function incorporated into a model, resulting in more

accurate results compared to those using the original loss function alone.

Comment Reviewer 2: (general comment) For the readers’ advantage, they need to elaborate on the point description of the proposed FLC and BLC models along with better and more comprehensive/professional looking diagrams. In its present form, the paper looks like copy pasted from a scientific report or a thesis.

Response After getting feedback from the reviewer, I have redrawn it following the advice. It is redrawn in the section

in Model Designs in the subsection in FLC and BLC models.

Fig 10. Front-loaded connection model

Fig 11. Back-loaded connection model

Comment Specific Comment 2-1: In the title they mention the use of Deep NN, but in the model description they

mention CNN. Why they have not mentioned CNN in the title?

Response: I have edited my title according to the feedback from reviewers.

On the Data Shifting: Predicting PM2.5 levels by using Enhanced Deep neural network enhanced

based nural networkOn the Data Shifting: Predicting PM2.5 levels using CNN-based neural network

Comment Specific Comment 2-2: In the Model Design Section (FLC and BLC), they mentioned the conclusion

along with a description (although inadequate) of the models (in lines 320-322: The results clearly

demonstrate that both the front-loaded and back-loaded models proposed outperform other models,

and the wrapped model exhibits superior performance compared to the one without the wrapped loss

function). Is it required here?

Response:

---

## [Decision Letter · Decision Letter 1]

23 Aug 2024

PONE-D-24-05738R1On the Data Shifting: Predicting PM2.5 levels by using CNN based neural networkPLOS ONE

Dear Dr. Hossen,

Thank you for submitting your manuscript to PLOS ONE. After careful consideration, we feel that it has merit but does not fully meet PLOS ONE’s publication criteria as it currently stands. Therefore, we invite you to submit a revised version of the manuscript that addresses the points raised during the review process.

**ACADEMIC EDITOR: Major revisions**

We look forward to receiving your revised manuscript.

Kind regards,

Worradorn Phairuang, Ph.D.

Academic Editor

PLOS ONE

Reviewers' comments:

Reviewer's Responses to Questions

**Comments to the Author**

1. If the authors have adequately addressed your comments raised in a previous round of review and you feel that this manuscript is now acceptable for publication, you may indicate that here to bypass the “Comments to the Author” section, enter your conflict of interest statement in the “Confidential to Editor” section, and submit your "Accept" recommendation.

Reviewer #3: (No Response)

Reviewer #4: All comments have been addressed

2. Is the manuscript technically sound, and do the data support the conclusions?

Reviewer #3: Partly

Reviewer #4: Yes

3. Has the statistical analysis been performed appropriately and rigorously? 

Reviewer #3: No

Reviewer #4: Yes

4. Have the authors made all data underlying the findings in their manuscript fully available?

Reviewer #3: (No Response)

Reviewer #4: No

5. Is the manuscript presented in an intelligible fashion and written in standard English?

Reviewer #3: No

Reviewer #4: Yes

6. Review Comments to the Author

Reviewer #3: The paper still needs significant improvement because the current version does not clearly explain the proposed methods and results. My comments for improvement as follows,

1. Abstract is not clear about what the paper proposes.

2. The paper needs to clearly outline the main contributions are, such as analysis of data drifting, drift detection, deep learning model (FLC and BLC Models), or the application of wrapped loss function in the models.

3. The introduction should clearly and explicitly describe the problem statement and the contributions.

4. In Previous work section, the paper frequently uses the subject "authors". It is unclear whether this refers to the authors of the current paper or the authors cited in the references.

5. In lines 112-119 in page 4, the statement "Among them, CNN-based models stand out for their 115

ability to process historical data from EPA stations based on the spatial distribution of 116

the sites [28]" attempts to conclude that CNN is better based on [28]. Among various models for various data in refs [22-28], it would be better to clearly explain why CNN is preferred, supported by strong evidence.

6.The Previous work section does not discuss the attention-base model which is proposed by the paper.

7. From Fig 3 to Fig 8, it is unclear what the y-axis values represent. Are they the results from Eq (3), (4), (5) and (6) or just P-value? It is important to note that the interpretations of the results from PC differ those from other methods.

8. In page 12, it is better explain FLC and BLC models more detail.

9. It is strongly recommended to reorganize the paper.

Reviewer #4: (No Response)

7. PLOS authors have the option to publish the peer review history of their article (what does this mean?). If published, this will include your full peer review and any attached files.

Reviewer #3: No

Reviewer #4: **Yes: **Dr. Nishit Aman

---

## [Author Response · Author response to Decision Letter 1]

1 Sep 2024

Plos Reviewers Response

Comment 1: Abstract is not clear about what the paper proposes.

Response After getting the feedback,I have written it, and elaborated on the purposes of my work.

Comment 2. The paper needs to clearly outline the main contributions are, such as analysis of data drifting, drift

detection, deep learning model (FLC and BLC Models), or the application of wrapped loss function in

the models.

Response I have written the main contributions of the paper in the introduction section

Comment 3. The introduction should clearly and explicitly describe the problem statement and the contributions.

Response I have rewritten the problem statement and contribution clearly in the Introduction section.

Comment 4. In Previous work section, the paper frequently uses the subject "authors". It is unclear whether this

refers to the authors of the current paper or the authors cited in the references.

Response I have edited the authors from the paper and stated clearly with the author’s name of the paper.

Comment 5. In lines 112-119 in page 4, the statement "Among them, CNN-based models stand out for their 115

ability to process historical data from EPA stations based on the spatial distribution of 116 the sites

[28]" attempts to conclude that CNN is better based on [28]. Among various models for various data in

refs [22-28], it would be better to clearly explain why CNN is preferred, supported by strong evidence.

Response After getting feedback, I have included a few works where that applied CNN based model for PM2.5

predictions the lines are given below from 119-124.

Comment 6.The Previous work section does not discuss the attention-base model which is proposed by the paper. .

Response I have discussed the attention base model on the previous work section.

Comment 7. From Fig 3 to Fig 8, it is unclear what the y-axis values represent. Are they the results from Eq

(3), (4), (5), and (6) or just P-value? It is important to note that the interpretations of the results from

PC differ from those of other methods. From Fig 3 to Fig 8 the results are from equations 3-6, those

equations calculated the P value finally. The interpretation of those results is given there.

Response From the equation (3) Eq (3), (4), (5), and (6), calculated the P value and value are illustrated on Figure 3-8.

Heatmaps are based on the P value that we get after those equations are calculated.

Comment 8. In page 12, it is better to explain FLC and BLC models in more detail.

Response The model description, I have described those models in more detail in FLC and BLC models sections.

Comment 9. It is strongly recommended to reorganize the paper.

Response I organize the paper according to feedback.

11. Plos second Reviewers Response

Comment o Line 98-203: I think there is no need of this Section giving too much theoretical information. It could

be more appropriate for a review paper. It is better to shorten this part and include as part of the

introduction.

Response I have edited the section and shortened it as I needed to discuss the previous work so I kept it and the another

reviewer also asked to introduce the applied model in details..

Comment o Add a section on Study Area and Data. Give details about the study area with Map with location of

various monitoring stations.

Response I have added a section in the name Study area and data and drawn a map on the location and described it.

Comment o Include a Section on Methods giving full details in sequence about the method used in the study. A

typical machine learning related work should include section as Data collection and pre-processing,

EDA and Feature engineering, Model development and evaluation, and finally prediction.

Response I have discussed the data collection and analysis of how collected the data in Data collection and pre-processing

section . There are several section as Model development and prediction .

Comment o . In figures, the default design for axes, labels, and legend has been used. Make label bold and enlarge.

Response I have redrawn the figures and made them bold too accordingly.

Comment o Write which software packages or libraries were used for the statistical analysis. Give references too.

Response I have given the references for the work’s statistical analysis as there are three techniques so I gave all of

them references too.

Comment o In conclusion, add a few sentences on policy implications and future work. Add limitations here.

Response I have added future work and limitations too in the following paragraph.

---

## [Decision Letter · Decision Letter 2]

1 Nov 2024

PONE-D-24-05738R2On the Data Shifting: Predicting PM2.5 levels by using CNN based neural networkPLOS ONE

Dear Dr. Hossen,

Thank you for submitting your manuscript to PLOS ONE. After careful consideration, we feel that it has merit but does not fully meet PLOS ONE’s publication criteria as it currently stands. Therefore, we invite you to submit a revised version of the manuscript that addresses the points raised during the review process.

**ACADEMIC EDITOR: ** In an effort to provide a speedy and rigorous review, I am your new academic editor. Accordingly, I have reviewed your manuscript and the revisions provided by the reviewers. Beyond what the reviewers have already said, I would like for a few minor changes to be added to ensure your manuscript is easily understood, and located by the appropriate audience. Specifically, Please ensure your figures have sufficient description. The description for most figures is too short. You will also want to provide units for PM2.5, which I could not locate in figure captions or in many parts of the text. (i.e. MAE is in the units of PM2.5 concentration, but units are not listed).consider a revision to your manuscript title. As it is, the title does not inform the reader about all of the major work that you've completed and is not representative of the manuscript. Current title: "On the Data Shifting: Predicting PM2.5 levels by using CNN based neural network". My critique is that "data shifting" is too vague and you've done more than just use CNNs. You should revise the title, a potential recommendation could be something like "Improving PM2.5 pollution predictions by correcting for annual data drift with wrapped loss functions evaluated against CNNs and LSTMs". Please feel free to use the suggested or modify as you like.Please check over the text for readability and English in the manuscript. For example, the use of "yanking" in the abstract is confusing and that sentence should be reworked.Consider adding specific major metrics in your conclusion and abstract sections (i.e. how much did your wrapped loss function improve on the accuracy when dealing with data drift?); this will increase audience interest and usability of your findings.Ensure abbreviations and terminology are consistent throughout the manuscript (i.e. p-value vs P value).Lastly, keywords. Considering adding more if allowed. Additionally, consider related terms beyond abbreviations.

Upon completion, I will review your article as soon as it comes back in and provide you with a timely final decision. I will not send it back to reviewers.

We look forward to receiving your revised manuscript.

Kind regards,

Kristofer Lasko, PhD

Academic Editor

PLOS ONE

Journal Requirements:

Reviewers' comments:

Reviewer's Responses to Questions

**Comments to the Author**

1. If the authors have adequately addressed your comments raised in a previous round of review and you feel that this manuscript is now acceptable for publication, you may indicate that here to bypass the “Comments to the Author” section, enter your conflict of interest statement in the “Confidential to Editor” section, and submit your "Accept" recommendation.

Reviewer #4: All comments have been addressed

Reviewer #5: All comments have been addressed

2. Is the manuscript technically sound, and do the data support the conclusions?

Reviewer #4: Yes

Reviewer #5: Yes

3. Has the statistical analysis been performed appropriately and rigorously? 

Reviewer #4: Yes

Reviewer #5: Yes

4. Have the authors made all data underlying the findings in their manuscript fully available?

Reviewer #4: Yes

Reviewer #5: Yes

5. Is the manuscript presented in an intelligible fashion and written in standard English?

Reviewer #4: Yes

Reviewer #5: Yes

6. Review Comments to the Author

Reviewer #4: (No Response)

Reviewer #5: Two models that consider the characteristics of data drifting for PM2.5 prediction are proposed and all comments have been addressed.

7. PLOS authors have the option to publish the peer review history of their article (what does this mean?). If published, this will include your full peer review and any attached files.

Reviewer #4: **Yes: **Dr. Nishit Aman

Reviewer #5: No

---

## [Author Response · Author response to Decision Letter 2]

7 Nov 2024

Point-by-Point Responses and the Reviewer Comments of

"Enhancing PM2.5 Prediction by Mitigating Annual Data Drift

Using Wrapped Loss and Neural Networks" by

Md Khalid Hossen*, Yan-Tsung Peng, Meng Chang Chen

Comment 1: Please ensure your figures have sufficient description. The description for most figures is too short.

You will also want to provide units for PM2.5, which I could not locate in figure captions or in many

parts of the text. (i.e. MAE is in the units of PM2.5 concentration, but units are not listed)

Response After getting the feedback, I have included the units for MAE (μg/m3), PM2.5(μg/m3) and unit for RMSE

(μg/m3) too. I have included in the text and in their figure too.

Comment 2. consider a revision to your manuscript title. As it is, the title does not inform the reader about all

of the major work that you’ve completed and is not representative of the manuscript. Current title:

"On the Data Shifting: Predicting PM2.5 levels by using CNN based neural network". My critique

is that "data shifting" is too vague and you’ve done more than just use CNNs. You should revise the

title, a potential recommendation could be something like "Improving PM2.5 pollution predictions by

correcting for annual data drift with wrapped loss functions evaluated against CNNs and LSTMs".

Please feel free to use the suggested or modify as you like.

Response Thank you for your suggestion and I modify a little .The current tile are given below "Enhancing PM2.5

Prediction by Mitigating Annual Data Drift Using Wrapped Loss and Neural Networks".

Comment 3. Please check over the text for readability and English in the manuscript. For example, the use of

"yanking" in the abstract is confusing and that sentence should be reworked.

Response I have revoked the sentence. I have checked the text and readibilty and modified accordingly.

Comment 4. Consider adding specific major metrics in your conclusion and abstract sections (i.e. how much did

your wrapped loss function improve on the accuracy when dealing with data drift?); this will increase

audience interest and usability of your findings.

Response Thank you for your suggestions. I have included the model performance percentage how much they improved

it and included in the Abstract and conclusion.

Comment 5. Ensure abbreviations and terminology are consistent throughout the manuscript (i.e. p-value vs P

value).

Response After getting feedback, I have fixed the manuscript only P value .

Comment 6. Lastly, keywords. Considering adding more if allowed. Additionally, consider related terms beyond

abbreviations.

Response I have added more keyword .

---

## [Editor Report · Decision Letter 3]

11 Nov 2024

Enhancing PM2.5 Prediction by Mitigating Annual Data Drift

Using Wrapped Loss and Neural Networks

PONE-D-24-05738R3

Dear Dr. Hossen,

We’re pleased to inform you that your manuscript has been judged scientifically suitable for publication and will be formally accepted for publication once it meets all outstanding technical requirements.

Kind regards,

Kristofer Lasko, PhD

Academic Editor

PLOS ONE
---

## [Editor Report · Acceptance letter]

18 Nov 2024

PONE-D-24-05738R3 

PLOS ONE

Dear Dr. Hossen, 

I'm pleased to inform you that your manuscript has been deemed suitable for publication in PLOS ONE. Congratulations! Your manuscript is now being handed over to our production team.

Kind regards, 

on behalf of

Dr. Kristofer Lasko 

Academic Editor

PLOS ONE